# Strength in numbers: optimal and scalable combination of LHC new-physics searches

Jack Y. Araz[a], Andy Buckley[b], Benjamin Fuks[c], Humberto Reyes-Gonzalez[d,e],
Wolfgang Waltenberger[f,g], Sophie L. Williamson[h], Jamie Yellen[b]

**a** *Institute for Particle Physics Phenomenology, Durham University, Durham, UK*

**b** *School of Physics and Astronomy, University of Glasgow, Glasgow, United Kingdom*

**c** *Laboratoire de Physique Théorique et Hautes Énergies (LPTHE), UMR 7589, Sorbonne Université et CNRS, 4 place Jussieu, 75252 Paris Cedex 05, France*

**d** *Department of Physics, University of Genova, Via Dodecaneso 33, 16146 Genova, Italy*

**e** *INFN, Sezione di Genova, Via Dodenasco 33, I-16146 Genova, Italy*

**f** *Institut für Hochenergiephysik, Österreichische Akademie der Wissenschaften, Nikolsdorfer Gasse 18, 1050 Wien, Austria*

**g** *University of Vienna, Faculty of Physics, Boltzmanngasse 5, A-1090 Wien, Austria*

**h** *Institute for Theoretical Physics, Karlsruhe Institute of Technology, 76128 Karlsruhe, Germany*

## Abstract

To gain a comprehensive view of what the LHC tells us about physics beyond the Standard Model (BSM), it is crucial that different BSM-sensitive analyses can be combined. But in general search-analyses are not statistically orthogonal, so performing comprehensive combinations requires knowledge of the extent to which the same events co-populate multiple analyses' signal regions. We present a novel, stochastic method to determine this degree of overlap, and a graph algorithm to efficiently find the combination of signal regions with no mutual overlap that optimises expected upper limits on BSM-model cross-sections. The gain in exclusion power relative to single-analysis limits is demonstrated with models with varying degrees of complexity, ranging from simplified models to a 19-dimensional supersymmetric model.

# 1 Introduction

The ATLAS and CMS experiments at the Large Hadron Collider (LHC) are performing direct searches for new physics beyond the Standard Model (BSM) in many different channels. The previous decade of LHC operation has already put strong constaints on the most obvious models of BSM physics, pushing their viable configurations to arguably untenable high new-particle masses; compared to these simple models pushed to extreme configurations, it is natural also for models with subtler phenomenology to enter the new-physics discourse. Such models bring increasing complexity in both the dimensionality of their parameter spaces, and the range of phenomenology possible within them. This leads to an increasing presumption that new physics will not be discoverable via a single, powerful experimental signature, but will disperse across many signatures at a level below direct exclusion in any one search analysis.

These factors cause a major logistical headache for LHC data-interpretation. At the fully detector-simulated level used for experiment interpretations, adaptive samplings and scans of high-dimensional spaces are not feasible, yet the few-parameter simplified-model approach used in the first LHC runs will no longer be representative of the analysis power to constrain actually viable BSM models. In this mode, it is clear that analyses must be systematically combined together, and initial scoping of viable parameter-space regions performed via a more lightweight approximation of experiment response, such as via the MADANALYSIS 5 [1],

CHECKMATE [2], RIVET + CONTUR [3, 4], GAMBIT [5], or similar toolkits.

In this paper we focus on the first problem: how to best combine analyses for optimal statistical significance, which for the purposes of our analysis is the ability to exclude a specific BSM model point at fixed confidence level. Definitive LHC statements about any dispersed signature will require combination of as many analyses as possible, but not all analyses *can* be combined. Were we simply to combine the test statistics of every signal region (SR) from every analysis available in the public collections, we would certainly double-count physics effects, since the same events will manage to pass multiple analyses' event-selection cuts and observable binnings.

Different sets of observables are used for selection-cut purposes in each analysis, but the disjoint choices are typically highly correlated through a complex dependence on the rest of the selection phase-space. It is hence impossible to reliably identify degrees of overlap directly from a list of cut observables and values. And even when the analysis overlaps *are* known, there remains the problem of identifying which compatible subset will place the optimal constraints on any given BSM model.

Approaches so far have hence been manual, and rather conservative [4, 6]. To scale up to the full set of LHC legacy analyses at 7, 13, and 13.6 TeV, and to obtain maximal statistical limits from the resulting combination, a more quantitative and automated approach is needed. This paper provides a blueprint for such an approach: we use the MADANALYSIS 5 analysis toolkit in conjunction with SMODELS [7] to estimate degrees of analysis overlap over hundreds of signal regions, and propose a new graph-based algorithm to optimise the subset of non-overlapping analyses used for testing a given BSM model.

In this work we determine the parameter space accessed by the topologies which populate the different signal regions contained in a sample of 18 CMS and ATLAS analyses, the current maximum overlap between the MADANALYSIS 5 and SMODELS re-interpretation toolkits. While far from a complete set, this is sufficient to illustrate the complexity of (undeclared) overlapping SR acceptances, and the non-triviality of identifying the most significantly exclusionary, combinable subset of SRs for a given BSM model.

In Section 2, by sampling over the minimal parameter spaces of the SMODELS simplified-model topologies able to populate these SRs, and using a version of MADANALYSIS 5 modified to provide information for per-event Poisson bootstrapping, we estimate the statistical overlaps between the resulting set of 355 signal regions. Applying a threshold on the degree of overlap acceptable in combination then results in a matrix of acceptable SR–SR combinations, from which the space of optimal subsets can be explored.

In Section 3, we find that a powerful method for doing so is to represent the SRs in a graph-theoretic form, in which sensitivity maximisation for a variety of physics-performance metrics can be formalised as a weighted longest-path problem.

In Section 4, we apply this technique on a series of increasingly complex and general BSM models, ranging from "closure tests" on simplified models, to compact models with dispersed phenomenology, to the 19-dimensional phenomenological Minimal Supersymmetric Standard Model (pMSSM-19).

We conclude with reflections on what is needed in technical and community-coordination terms to bring this method and the resulting gains in LHC physics sensitivity to practical realisation.

# 2 Overlap estimation

To investigate how analyses can be combined to provide the most stringent constraints on a BSM model point, we choose the selection of analyses available both in SModelS and MadAnalysis 5 as our database. At the time of writing this includes 18 analyses: ATLAS-SUSY-2013-02 [8], ATLAS-SUSY-2013-04 [9], ATLAS-SUSY-2013-05 [10], ATLAS-SUSY-2013-11 [11], ATLAS-SUSY-2013-21 [12], ATLAS-SUSY-2015-06 [13], ATLAS-SUSY-2016-07 [14], ATLAS-SUSY-2018-04 [15], ATLAS-SUSY-2018-06 [16], ATLAS-SUSY-2018-31 [17], ATLAS-SUSY-2018-32 [18], ATLAS-SUSY-2019-08 [19]; CMS-SUS-13-011 [20], CMS-SUS-13-012 [21], CMS-SUS-16-033 [22], CMS-SUS-16-039 [23], CMS-SUS-16-048 [24]. CMS-SUS-17-001 [25], and CMS-SUS-19-006 [26].

The cascade decays, or *topologies*, covered by these analyses are simplified so that they focus on the production of two massive BSM states that each decay to at most 2–3 final-state particles. The topologies covered by these analyses, using the SModelS naming convention [27], are: T1, T1bbbb, T1btbt, T1tttt(-off), T2, T2bb, T2tt(-off), T2bbWW(-off), T2bt, T2cc, T3GQ, T5, T5bbbb, T5tctc, T5tttt, T5GQ, T5WW(-off), T5WZh, T5ZZ, T6bbhh, T6bbWW(-off), T6WW(-off), T6WZh, TChiChipmSlepL, TChiChipmSlepStau, TChiChipmStauStau, TChiChipmSlepSlep, TChipChimSlepSnu, TSlepSlep, TChiZZ TChiWH, TChiWW, TChiWZ(-off), TChiZoff, TGQ, TSlepSlep, and TStauStau.

## 2.1 Model-space sampling and event generation

In order to obtain robust conclusions about potential signal overlaps for arbitrary scenarios, we proceed as follows. For each analysis, we construct a convex hull in each simplified model's parameter space that is accessed by a given topology, carried out using the efficiency maps implemented in SModelS [28]. The efficiency maps give upper limits on the production cross-sections of the two relevant BSM states, and depend on the masses in the simplified decay chains. For each simplified model, one such convex hull exists for each analysis that has a result for that given simplified model. We are interested in the joint set of convex hulls corresponding to each simplified model. Thus, we construct a contour around the mass-parameter space beyond which the expected event-yield from all corresponding analyses is zero. In this way, the union of regions will be populated with events, without multiply populating those shared between analyses. We uniformly generate events within this joint convex hulls, so to only introduce an uninformative flat prior in our procedure.

The MC events were generated at LO with MadGraph5_aMC@NLO v2.6.5 [29] at the partonic level with the NNPDF 2.3 LO [30] set of parton distribution functions via the Lhapdf library [31], with parton-showering and hadronisation simulated by Pythia 8 [32] through the MadGraph5_aMC@NLO interface. Detector-level events were obtained with Delphes 3 and FastJet [33,34], executed through MadAnalysis 5 with analysis-specific configurations interleaved with the event-selection logic. The input for the generation pipeline was a corresponding SLHA-format [35] data-file for each topology, with the masses of the produced, final, and (in some cases) intermediate BSM states defined as free parameters. The initial partonic processes in the generation chain were in all cases direct production of the topology's massive BSM states, with decay chains implemented via Pythia 8's decay mechanism.

The required output of the MC generation procedure is a binary *acceptance matrix* $\Theta$ of shape $N_{\text{evt}} \times N_{\text{SR}}$, where $\Theta_{e,s} = 1$ means that event $e$ populated SR-bin $s$, and *vice versa*

$\Theta_{e,s} = 0$ when event $e$ did not pass the cuts for SR $s$.[1] This matrix is produced using the new

```
set main.recast.TACO_output = <file-name>
```

command in MADANALYSIS 5, added to the framework for this purpose. The *acceptance matrices* are emitted as text files with each event corresponding to a pair of lines encoding first the list of floating-point event weights [36] (in this study we use only the nominal weight), and then a list of 0 and 1 characters corresponding to the $N_{\mathrm{SR}}$ signal regions. These files are written separately for each DELPHES 3 configuration to the location

```
<Output>/SAF/defaultset/<delphes-card-name>.<file-name>
```

in the output directory of the recasting process.

To determine the minimal number of Monte-Carlo events needed for a reliable estimation of the overlap matrix, we start with 100 events for 1000 random parameter points sampled from the union of the convex hulls of the signal regions, so with an initial N=100 000 events. For any pair of signal regions $\mathrm{SR}_1$ and $\mathrm{SR}_2$ populated with $n_1$ and $n_2$ events respectively, we then determine the number $k$ of shared events. If $k > 100$, we have accumulated enough statistics, and proceed to the bootstrapping procedure. For $k \leq 100$, with $n \equiv n_1 + n_2$, we use the confidence interval construction by Clopper and Pearson [37] of the binomial distribution

$$\mathrm{B}(n,p) = \binom{n}{k} p^k (1-p)^{n-k} \tag{1}$$

where $p$ is a free parameter defined as the *probability of overlap*. In order to guarantee enough events for the case of a negligible overlap, we need to obtain a one-sided (upper) confidence interval for $p$ at confidence level $\mathrm{CL} \equiv 1 - \alpha = 0.95$ and guarantee that it is below a certain threshold. From the Clopper–Pearson construction, this is computed as the $1 - \alpha$ quantile of the $\beta$ distribution

$$f(p; k, n) = \beta_{1-\alpha; k+1; n-k}. \tag{2}$$

If this upper bound is below the arbitrarily chosen threshold, $f < 0.01$, we assume that we have accumulated enough statistics to safely infer the potential absence of a significant overlap, and we confidently proceed to the bootstrapping procedure.

Using these criteria we can employ the logic of Figure 1. It will guarantee that enough statistics is available to robustly and reliably determine both a significant or negligible overlap between a given pair of signal regions.

## 2.2 Overlap-matrix estimation

Once with a set of sufficiently populated SRs, we are ready to determine whether or not such SRs are approximately orthogonal with respect to one another.

The Pearson correlation can be estimated from the acceptance matrix via the event-averaged acceptance covariance,

$$\begin{aligned}
\mathrm{cov}_{ij} &= \langle \Theta_i \Theta_j \rangle - \langle \Theta_i \rangle \langle \Theta_j \rangle \\
&\equiv \frac{\sum_e \Theta_{e,i} \Theta_{e,j}}{N_{\mathrm{evt}}} - \frac{\sum_{e'} Y_{e',i} \cdot \sum_{e''} Y_{e'',j}}{N_{\mathrm{evt}}^2},
\end{aligned} \tag{3}$$

---

[1] In general this matrix need not be binary, and can represent a per-event bin yield, for observables that can have multiple fills per event. Event weighting also complicates matters. But in the current context of binary acceptance or rejection of unweighted events, $\Theta_{ij} \in \{0, 1\}$.

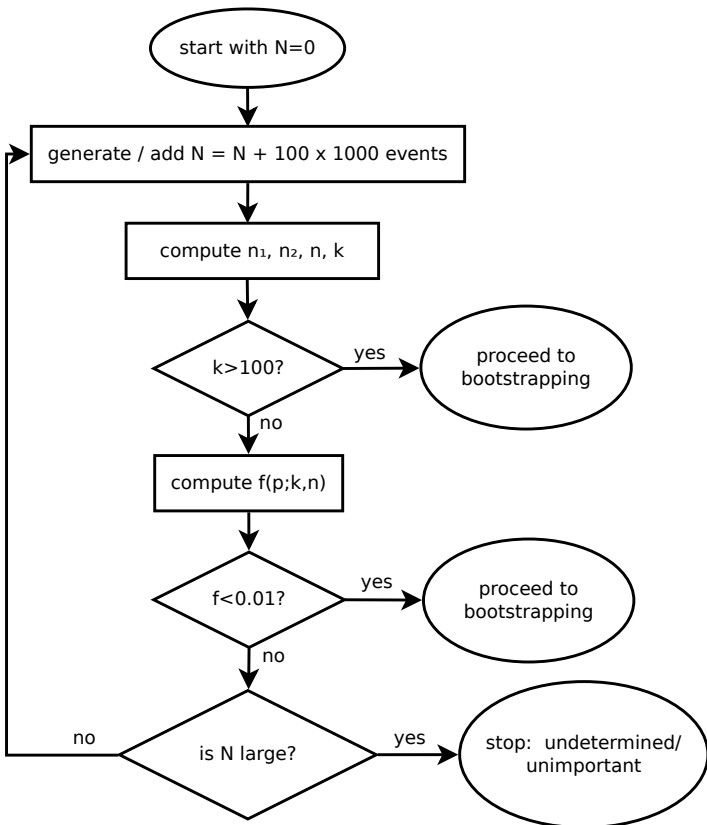

Figure 1: Flowchart for the determination of the number of Monte Carlo events needed for estimation of the overlap matrix.

where $N_{\text{evt}}$ is the number of events in the estimation sample and, as made explicit in the second line, $i$ and $j$ are SR indices. This method is possible because the entire event-wise acceptance matrix is available and hence overlaps can be estimated by averaging over the event axis of the matrix.

An equivalent approach, taken by the current code, is to perform bootstrap sampling from a unit Poisson distribution. Each event is assigned $N_{\text{boot}}$ random "bootstrap weights" $w_{e,b} \sim \text{Pois}(\lambda = 1)$, which are aggregated on to $N_{\text{boot}}$ replicas of the SRs' yield estimates. The result is a $N_{\text{SR}} \times N_{\text{boot}}$ bootstrapped *yield matrix* $Y$, which expresses the sum of event weights falling into the set of SRs for each of the $N_{\text{boot}}$ alternative histories generated from the single set of input events. The overlaps between SRs can then be determined from their common weight-fluctuations over the set of histories, i.e. another estimate of the covariance:

$$
\begin{aligned}
\text{cov}_{ij} &= \langle Y_i Y_j \rangle - \langle Y_i \rangle \langle Y_j \rangle \\
&\equiv \frac{\sum_b Y_{i,b} Y_{j,b}}{N_{\text{boot}}} - \frac{\sum_{b'} Y_{i,b'} \cdot \sum_{b''} Y_{j,b''}}{N_{\text{boot}}^2} \, .
\end{aligned}
\tag{4}
$$

The distinction is that the averaging is now over bootstrap replicas of the aggregate yields, rather than the per-event acceptance tuple. While not essential in the current implementation, the bootstrap approach avoids the need to manage a linearly growing acceptance matrix, in favour of a fixed-size $N_{\text{SR}} \times N_{\text{boot}}$ yield matrix, which may become computationally relevant for large event samples.

From the covariance matrix, obtained through either strategy, we define the *overlap matrix*

$$
\rho_{ij} = \frac{\text{cov}_{ij}}{\sqrt{\text{cov}_{ii} \, \text{cov}_{jj}}} \, ,
\tag{5}
$$

following the usual Pearson-correlation definition. Lower-triangle plots of this symmetric overlap matrix for the sets of signal regions common to SMODELS and MADANALYSIS 5 are shown in the appendix Figures 14 and 15 for 8 TeV and 13 TeV LHC data-analyses respectively, with patterns of highly and partially co-populated SRs clearly visible.

Finally, a binary *exclusivity matrix* $E$ between SR-pairs $\text{SR}_i$ and $\text{SR}_j$ is derived by applying an "acceptable overlap" threshold $T$ such that the exclusivity between SRs $i$ and $j$ is $E_{ij} = (|\rho_{ij}| \leq T)$. The value chosen for $T$ is at present somewhat subjective, reflecting that for each use-case there will be a finite value of $\rho_{ij}$ below which double-counting biases are not statistically resolvable: treating these low correlations as zero-correlations avoids blocking useful SR combinations due to irrelevant and noisy correlation estimates.

The procedure described above is implemented in the public PYTHON program TACO (Testing Analyses COrrelations), available at https://gitlab.com/t-a-c-o/taco_code.

## 3 Optimal signal-region combination

Armed with the exclusivity matrix from the previous section, for a choice of overlap threshold $T$, we now have the challenge of identifying the best-expected combination of SRs compatible with it. We consider this in two steps: first the combinatoric problem of efficiently constructing *all* allowed paths, and then the optimisations to this enabled by the specific definition of "best" used in BSM-analysis reinterpretation.

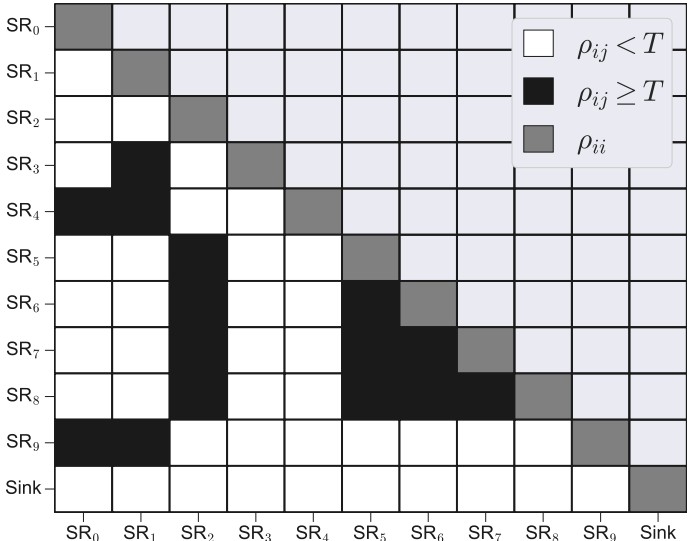

Figure 2: Overlap matrix of 10 signal regions $(SR_0 - SR_9)$ with values masked according to threshold $T$ giving the exclusivity matrix. The final *Sink* signal region has been inserted to provide a target for the path-finding algorithm

## 3.1 Compatible signal-region sets as path-finding

Without prior information about overlaps or statistical significances, the process of finding a preferred subset of signal regions from a set of size $n$ is a combinatorial challenge with $2^n$ possible solutions. Exhaustively generating and evaluating each such combination hence suffers from exponential time-complexity scaling, and is computationally impractical even for relatively small $n$, let alone the $n \sim \mathcal{O}(1000)$ required by real reinterpretations.

However, considering the SR-acceptance exclusivities $E_{ij}$, the majority of these $N$ combinations transpire to be *forbidden*, as large-$r$ naïve subsets become overwhelmingly likely to contain at least one overlapping pair of SRs. The question then becomes whether, given prior awareness of $E_{ij}$, it is possible to evaluate all *allowed* SR-combinations more efficiently than exhaustive generation followed by overlap-checking.

In this section we show that the answer to this question is yes, and that the problem can be usefully recast as finding an optimum path through a directed acyclic graph (DAG). We present an algorithm that reduces the asymptotic time complexity by efficiently selecting path-elements based on recursive application of the SR exclusivity matrix, rendering the combinatoric problem not just tractable but computationally fast.

The key insight is to avoid generating invalid SR-combinations at all: this can be achieved by generating the combinations directly from the overlap matrix. Hence we must restrict the generated subsets only to those for which $E_{ij} = 1$ for all distinct $i, j$ in the set. This condition requires that if a subset of all possible signal regions is built up iteratively, its $j$th element must have no significant overlap with all the previously selected elements $0...j - 1$.

Figure 2 shows the exclusivity matrix $E_{ij}$ of ten signal regions, computed as the overlap matrix masked with a threshold $T$. The matrix elements $\rho_{ij}$ that fall below $T$ are shown as white, and those that are above are shaded black. For reasons that will become clear later, we restrict ourselves to constructing combinations by adding SRs to the subsets in

strictly increasing index order. Starting in the top left-hand corner of Figure 2 (at element $\rho_{00}$, or $(\mathrm{SR}_0, \mathrm{SR}_0)$), the signal regions available for combination with $\mathrm{SR}_0$ are limited to those corresponding to the white elements in the first column, *i.e.* $E_{i,0} = 1$. We define $A_i$ as the ordered set of all *allowed* (non-overlapping) SRs indices with respect to SR $_i$ SR indices such that

$$A_i \equiv \{j : \rho_{i,j} < T,\ i < j < n\}. \tag{6}$$

Using Figure 2 as an example, $A_0$ would be $\{1, 2, 3, 5, 6, 7, 8\}$. We can now expand on the previous definition of $K$, specifying $K_i$ as a set of all allowed paths with initial elements $\mathrm{SR}_i$ such that:

$$K_i \equiv \{\{\mathrm{SR}_i, \dots, \mathrm{SR}_{\mathrm{final}}\}, \dots\}, \tag{7}$$

In this construction $K_{i,j}$ would be the $j$th path within $K_i$, and by extension $K_{i,j,k}$ would refer to the $k$th element of $K_{i,j}$. Applying this formulism to Figure 2 and initiating a subset $K_{0,0,0}$ with $\mathrm{SR}_0$, the available options for the second element are given by indices in $A_0$ (equation (6)). It follows that $K_{0,0,1} = \mathrm{SR}_1$ as this is the first index in $A_0$, and thus $K_{0,0,2} = \mathrm{SR}_2$ as this is the first available SR-index that is allowed by the intersection of $A_0$ and $A_1$. Repeating the procedure and taking the intersection of $A_1$ and $A_2$ results in an empty set, meaning that $K_{0,0}$ is a complete subset of three signal regions with overlaps below $T$. The next combination, $K_{0,1}$, is the first allowed alternative to the final element of $K_{0,0}$: $\{\mathrm{SR}_0, \mathrm{SR}_1, \mathrm{SR}_2\}$ becomes $\{\mathrm{SR}_0, \mathrm{SR}_1, \mathrm{SR}_5\}$.

This method of building paths is close to that of a depth-first search through an unweighted directed acyclic graph where the "nodes" correspond to signal-regions and "edges" to the allowed pairwise SR-combinations. The directed and acyclic nature of the graph is enforced by the ordering of SRs and the edges always pointing from lower to higher indices. However, there is a major difference in that the choice of each signal region is dependent on those allowed by all previous signal regions in the path, or in other words the allowed vertices would be inherited. Fortunately this *hereditary condition* can be easily inserted into established DAG "simple path" algorithms [38, 39].

Recasting the problem as an optimum-path search requires a few minor changes to the definitions covered so far. Firstly, each path has to be defined between two points: a source and a sink. As previously stated, each combination within the subset $K_i$ has a defined source, however, the final signal region will depend on the path taken. A convenient way of dealing with this condition is to define a universally allowed $n$th signal region such that every possible path terminates at index $n$. This can be done by appending an $n$th "sink" signal region to $\rho$, this is shown in Figure 2 but can also be expressed as

$$\rho_{n,i} = \rho_{i,n} = 0.0\ :\ 0 \leq i \leq n. \tag{8}$$

This modification of $\rho$ necessitates that the definition of $A_i$ also be modified to include the $n$th term:

$$A_i \equiv \{j : \rho_{i,j} < T,\ i \leq j \leq n\}. \tag{9}$$

With $A_i$ defined in terms inclusive of $n$, we can define a modified hereditary depth-first search (HDFS) algorithm that generates all the available paths starting from an initial signal region. This algorithm proceeds by recursively appending diminishing subsets of allowed SRs $\mathcal{S}$, with the current subset $\mathcal{S}_c$ defined as the the intersection of $A_c$ with the previous subset such that

$$\mathcal{S}_c \equiv A_c \cap \mathcal{S}_{c-1}. \tag{10}$$

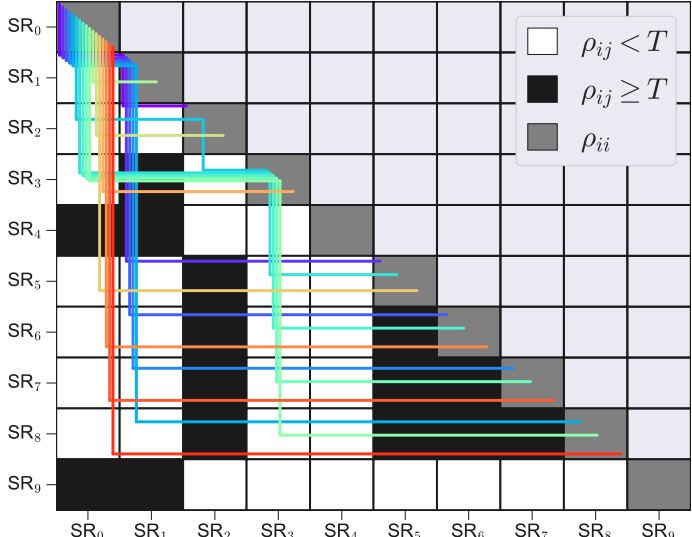

Figure 3: Exclusivity matrix of 10 signal regions ($SR_0 - SR_9$) with values masked according to threshold $T$. For clarity, the sink SR is not shown. The coloured lines show all allowed paths originating at $SR_0$.

The remaining compatible SRs are hence given by the total intersection of the compatible-SR sets for the elements already in the path. As this is constructed iteratively, each stage of completion-refinement needs only to be compared against the set of completions for the current final element, $S_{c-1}$). The HDFS algorithm uses this condition to efficiently exclude overlapping SR-combinations from consideration.

In summary, initiated from a source $SR_i$, with the first element of $S$ being $A_i$, the set of all allowed paths $K_i$ can be built by recursively evaluating the subsets of $S$. Once the current iteration has reached the "sink" $SR_n$, a full path is defined by the steps taken. Figure 3 shows the results from running this algorithm using the exclusivity matrix from Figure 2, for paths starting from $SR_0$. The full DAG HDFS algorithm is given in pseudocode as Algorithm 1 in Appendix B.

## 3.2 Weighted edges for sensitivity optimisation

In generating the set of allowed paths, we have been concerned only with SR-exclusivity and treated all graph edges (and hence SRs) as of equivalent value, within the fixed DAG ordering provided. But in our physics application, of course, this is not the case: for each specific BSM model, some signal regions will be more sensitive than others. For example, leptophilic models naturally tend to see most sensitivity in SRs with multilepton signatures; models with enhanced couplings to the third generation have most impact on $t$- and $b$-quark and $\tau$-lepton signatures; and dark-matter models favour jet + missing transverse-energy signatures. In addition, when not all SRs have the same integrated luminosity, SRs in high-luminosity datasets are naturally more sensitive than those in low-statistics ones. These intuitive sensitivity metrics can be incorporated into the graph model in the form of variable edge-weights.

Such weights should be motivated by the statistical goal being tested, and ideally should be additive so standard longest-path optimisation can be used to identify the most sensitive

allowed SR-combination. A typically appropriate choice for the edge weights, and the one used in this paper, is the logarithm of the expected likelihood-ratios (LLR) between the signal-model under test and the background-only model, $\ln(L_{\mathrm{sb}}/L_{\mathrm{b}})$, for pseudodata equal to the expected yields under the background-only model. This is motivated by the following logic:

1. As we are combining a set of direct-search analyses in which no individual significant signal was found, we choose to frame our mission primarily as maximising the volume of model-exclusion rather than a discovery. Our null hypothesis is hence the BSM signal model, and we seek to overturn it with a preference for the SM at every point in its parameter-space.

2. We hence aim to maximise the expected significance of exclusion $Z$ at each point in the BSM parameter space. Under the assumptions of Wald's Theorem [40], the expected significance is given by the square-root of the LLR between the models, hence maximising the LLR maximises the expected model-exclusion.

As any generated path is by definition composed of signal regions which can be treated as non-overlapping, the total log likelihood-ratio (LLR) $\sum_{i \in \mathrm{SRs}} \ln(L_{\mathrm{sb},i}/L_{\mathrm{b},i})$ of an SR subset is just the sum of such weights along its corresponding path candidate. The use of expected background pseudodata rather than the actual observed data-counts is important to avoid cherry-picking of statistical fluctuations: we identify the optimal SR-combinations for each point as if the data has not yet been recorded, to avoid bias. Other use-cases, in particular anomaly-detection, in which the observed data is compared to background expectations in search of the most consistent, discrepant non-overlapping subset of measurements, require a modified metric but with similar motivation. For such use cases, however, the edge-weights are in general no longer additive, resulting in a more complicated and CPU-intensive task.

In general, the optimal path can be found in reasonable time by evaluating the overall sensitivity metric for every allowed SR-combination identified by the HDFS algorithm of the previous section. However, in the case of additive weights, further algorithmic optimisations are possible by a) ordering the SRs in decreasing order of individual sensitivity, and b) exiting early from generation of allowed-path subsets for which there is no possibility of exceeding the metric obtained for the current maximum-sensitivity path. The first of these conditions is simply implemented by *a priori* ordering the SRs according to decreasing expected LLR, such that paths containing the expected dominant contributions to total LLR are evaluated first — this opens the possibility of evaluating only the sets of paths starting with the first $\mathcal{O}(10)$ SRs. The second, however, makes such a manual cutoff largely redundant by maintaining records of the highest complete-path LLR, and the sum of LLRs over all remaining SRs in $\mathcal{S}_c$ as the allowed paths are generated. Should the sum of the current path's LLR and its maximum possible completion become smaller than the current best complete path, there is no point in continuing to evaluate that set of completions and they can be "short-circuited" to further reduce the algorithmic complexity of the path-finding.

We refer to this combination of DAG hereditary depth-first search and these optimisations for weighted graph edges as the weighted HDFS algorithm (WHDFS). This algorithm is our final method for efficiently addressing the specific problem of finding the combination of statistically non-overlapping SRs which maximises their additive combination of expected LLR sensitivities to a given BSM model.

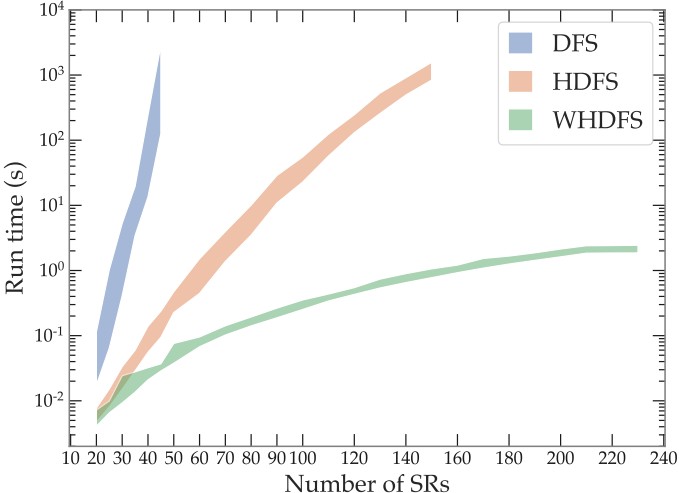

Figure 4: Comparison of CPU-runtime scaling against number of signal regions between the standard depth-first search (DFS), the hereditary DFS (HDFS), and the weighted hereditary DFS (WHDFS) algorithms.

## 3.3 Performance

The algorithm performance was evaluated by randomly selecting 20 mass-points from the "T1" simplified gluino-pair analysis to be shown in the following physics-results section, and calculating the optimal combinations for each of a set of reduced SR collections $\{SR_0, \ldots, SR_{m-1}\}$ and its corresponding $m \times m$ exclusivity submatrix. The number of elements $m \leq n$ in the reduced SR-sets was evaluated from $m = 20$ to 80 in steps of 10, and from $m = 80$ to 140 in steps of 20. The upper limit on $m$ was determined by the requirement to find 20 mass points (for timing-uncertainty estimation) with at least that many supported SRs.

Figure 4 shows the CPU-performance comparison between the three graph-based algorithms discussed in Section 3.1 on these SR-combination problems of varying size. The plot clearly shows that the simple depth-first search (DFS) does not scale sufficiently for physics purposes: in BSM scans considering many thousands or millions of model points and hundreds of SRs, the decision of which SR combination to use for each point needs to be made typically on the order of seconds, but the DFS algorithm requires hundreds of seconds by 40 SRs, with extremely strong exponential scaling. The HDFS algorithm fares much better, scaling up to 100 SRs with a flatter exponential growth than DFS, and with slightly sub-exponential thereafter. Regardless, it requires $\mathcal{O}(100)$ seconds for 100 SRs, insufficient for many practical applications. But the further optimisations enabled by the WHDFS formulation show a flatter still scaling exponent, with sub-exponential growth that becomes particularly flat for large SR counts. 230 SRs were obtained in around 2 seconds, very compatible with adaptive sampling, and indicating little issue in scaling further toward thousands of SRs. These performance gains indicate the effectiveness of the WHDFS algorithm and that it can meet the current practical requirements of large-scale analysis combination.

# 4 Results

To illustrate the power of our approach, we now present physics results for various BSM-reinterpretation scenarios. In order of increasing complexity, we first demonstrate increases in model-exclusion limits in the context of simplified models in Section 4.1. Raising the stakes, we then demonstrate the effect of our combinations on the pMSSM-19 model in Section 4.2. We end with a discussion of our combination results in the context of a simple $t$-channel dark matter model, this time fully recasting the relevant analyses, in Section 4.3.

As is the case throughout this paper, control regions are ignored, as are overlaps in the background expectations of the signal regions. This reflects an implicit assumption that the signal regions are specific enough to event topology and kinematic phase-space that events falling into them are indistinguishable between signal and background (the job of removing reducible backgrounds having already been performed by the pre-selection and SR-cut definitions). One could hence also perform the overlap estimation using large background-event samples in place of the sampling over signal models.

## 4.1 T1 simplified-model combination

Following the method described in Section 2, an overlap matrix was constructed from the selection of analyses available in SModelS and MadAnalysis 5. Using the SModelS database, an exclusivity matrix of 393 SRs was created with a 1% threshold of maximum overlap. The first test of the TACO formalism was to compare the combination results to the validation plots for analyses and topologies available in the SModelS database, chosen as this checks for consistency within a model-space completely understood and mapped by SModelS. The first simplified model chosen was the "T1" topology, which is a simplified version of gluino pair production in which each gluino undergoes a three-body decay $\tilde{g} \to q\bar{q}\tilde{\chi}_1^0$ to a light-flavor quark-antiquark pair plus the lightest stable particle (LSP) $\tilde{\chi}_1^0$.

Figure 5 shows the T1-topology validation plot of (a) expected and (b) observed limits for the combined SRs against three individual analyses. The contour lines show the exclusion limits in terms of the ratio of the predicted cross-section $\sigma_{\mathrm{pred}}$ and the upper limit on that cross-section $\sigma_{\mathrm{UL}}$, $r \equiv \sigma_{\mathrm{pred}}/\sigma_{\mathrm{UL}}$, such that $r = 1$ corresponds to the line of exclusion at 95% confidence. A total of 265 SRs were available with contributions to the T1 topology. For each point in the model space, the number of available SRs was determined by identifying those with efficiency maps whose parameter ranges included the model point. Once a set of available SRs was identified, they were ordered by the expected upper-limit (UL) on the expected yield (luminosity × cross-section × efficiency) at the model point for each SR. This selection and ordering of the signal regions was propagated to the exclusivity matrix, and the WHDFS SR-selection algorithm was applied. Figure 5 shows that the combined result pushes the exclusion line beyond that of the best performing analysis available in the current SModelS database by approximately 150 GeV.

Looking deeper into the combined results, Figure 6(a) shows the distribution of starting (*i.e.* lowest) SR-indices over the set of maximum-sensitivity combinations. This confirms the statements made in Section 3 that the efficiency of the path-finding would be greatly increased by sorting the exclusivity matrix by individual SR sensitivities. The histogram shows that, when ordered, the optimum combination is typically seen early in the iteration process, allowing many later path-sets to be vetoed when there is no prospect of their completions beating the current best. The right-hand plot of Figure 6 shows the percentage prevalence of

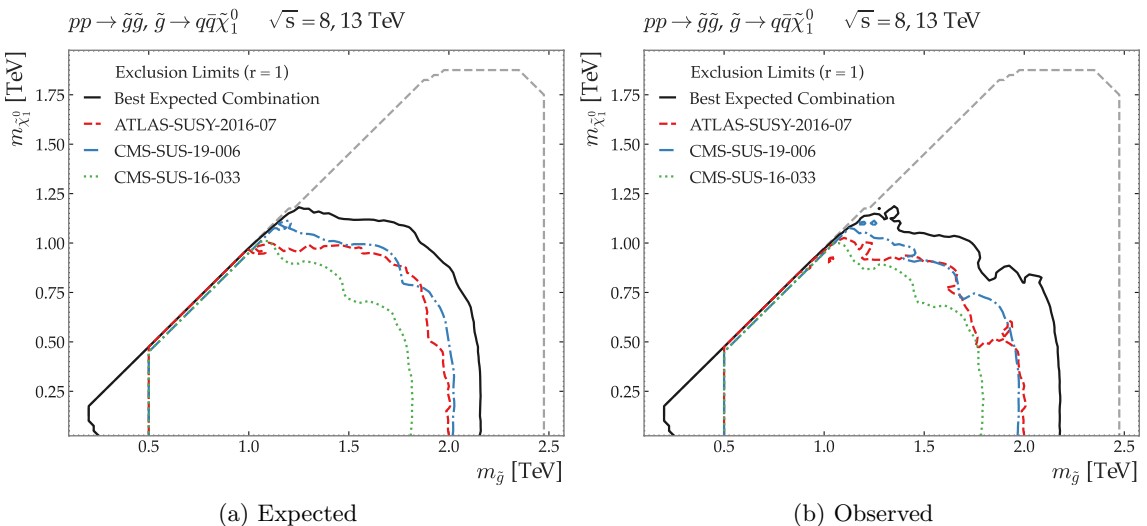

(a) Expected                                    (b) Observed

Figure 5: Validation plots comparing the (a) expected and (b) observed results of the TACO SR-combination against three individual-analysis limits available in the SMODELS results database for the T1 topology. The lines show the exclusion limits in terms of $r$-value where $r = 1$ is analogous to the 95% confidence-level exclusion. The CMS-SUS-19-006-ma5 analysis is so-named due to the efficiency map being obtained for SMODELS by use of MADANALYSIS 5 rather than direct from the experiment-provided analysis data. The dashed-grey line indicates the boundary of the efficiency maps.

the number of SRs in each best-sensitivity combination, with typically 6–10 of the available 265 SRs being used. This small number is conveniently compatible with expensive statistical treatments such as coherent profiling or marginalisation of systematic uncertainties across analyses, which would be prohibitively expensive over the 265 (and ever-increasing) full set of SRs.

We also considered the T1tttt topology, a modification of the T1 model in which the gluino decays exclusively into top quark–antiquark pairs ($\tilde{g} \to t\bar{t}\tilde{\chi}_1^0$). Figure 7 shows (a) the T1tttt topology validation plot of expected and (b) observed results, again comparing the exclusion ranges of the combined SRs to that of three individual analyses. The construction of the plot follows the same methods used for Figure 5. Similarly, the dominant contribution to the T1tttt combinations is the CMS-SUS-19-006 analysis [26] (seen in the blue dot-dashed exclusion line). Again, a significant expansion of the 95% exclusion contour over the single-SR results is seen, with the combination seen to smooth out the particular weakness of the most constraining analysis around $m_{\tilde{\chi}_1^0} \sim 1.1 \, \text{TeV}$.

## 4.2  pMSSM-19 reinterpretation

With the machinery in place to construct the exclusivity matrix from SRs based on a given model-point, it was now possible to extend the analysis to increasingly complex models. The 19-parameter phenomenological Minimal Supersymmetric Standard Model (pMSSM-19) was chosen as a testing ground for reinterpretation, as a model with considerably more degrees of freedom than typically studied in experimental publications. We used data points sourced from

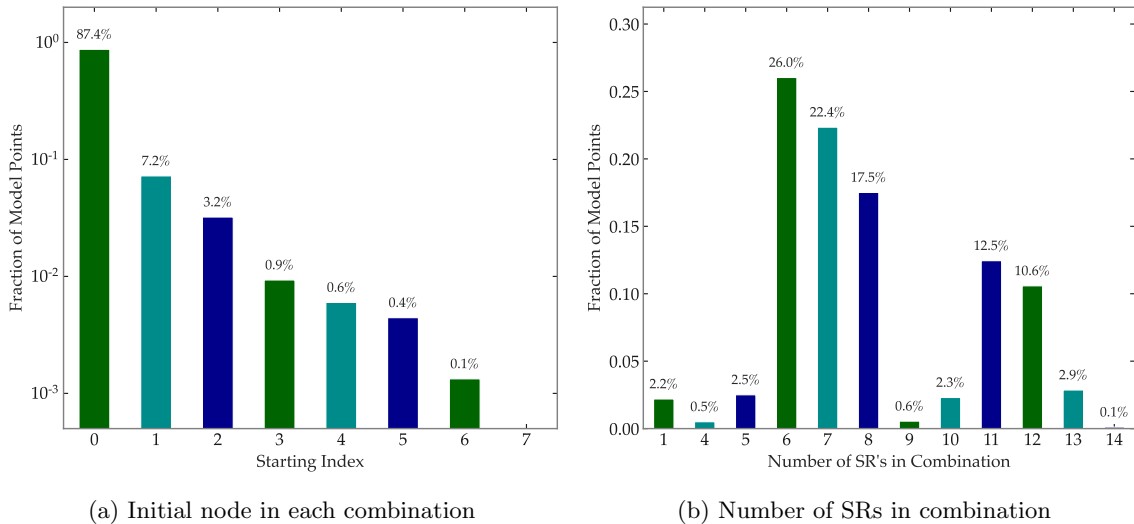

(a) Initial node in each combination          (b) Number of SRs in combination

Figure 6: Fractional distributions of (a) starting SR-index and (b) number of SRs in each combination. Data is taken from the optimum SR combination found for each model point in the T1 model space. As mentioned in Section 3 the nodes are constructed from an ordered set of SRs that is optimised for each point in the model space.

the ATLAS 2015 pMSSM-19 scan paper [41], independently for that paper's two scenarios of bino-like and wino-like LSPs. These points were identified by ATLAS according to their model viability compared to 8 TeV ATLAS data, and hence provide an *a priori* interesting set for re-evaluation against 13 TeV LHC data.

$p$-values from 13 TeV LHC Run-2 data analyses were calculated from the first (randomly ordered) 27 000 model points in the bino and wino scenarios separately, run through the SMODELS analysis chain and discarding those points which lay outside the bounds of the SMODELS efficiency maps, with ∼ 20 000 points surviving in each run.

LLRs were calculated using the SMODELS best-single-expected SR-selection process and the best-expected-combination results. The resulting $p$-value distributions from the pMSSM-19 bino-LSP reinterpretation are shown in Figure 8. The histograms show that in both the (a) expected and (b) observed cases the combination procedure moves a significant fraction of points from below the 95% exclusion into the excluded category, an increase in exclusion fraction from approximately 35% to 70% of all points in the ATLAS set of bino-like models.

This mean shift from single to combined could be seen as confirmation of the exclusionary power of the TACO approach, but before making any conclusions it is prudent to review exactly how the model points are behaving on a bin-to-bin level. To this end, Figure 9 shows the transition matrices (also known as stochastic matrices) for the pMSSM-19 bino dataset, showing the probability of a model point "moving" between $p$-value bins depending on whether the single-SR or combined-SR LLR construction method is used. This can either be framed as the probability of points in a particular single-SR bin moving to each combined-SR bin, or the "origin distribution" of the points ending up in a particular combined-SR $p$-value bin; both versions are informative and are shown in the left- and right-hand subfigure columns respectively, with expected and observed results in the subfigure rows. As the values in each matrix are given as a probability $P(\text{row}|\text{column})$ the sum over the column values equals 1.

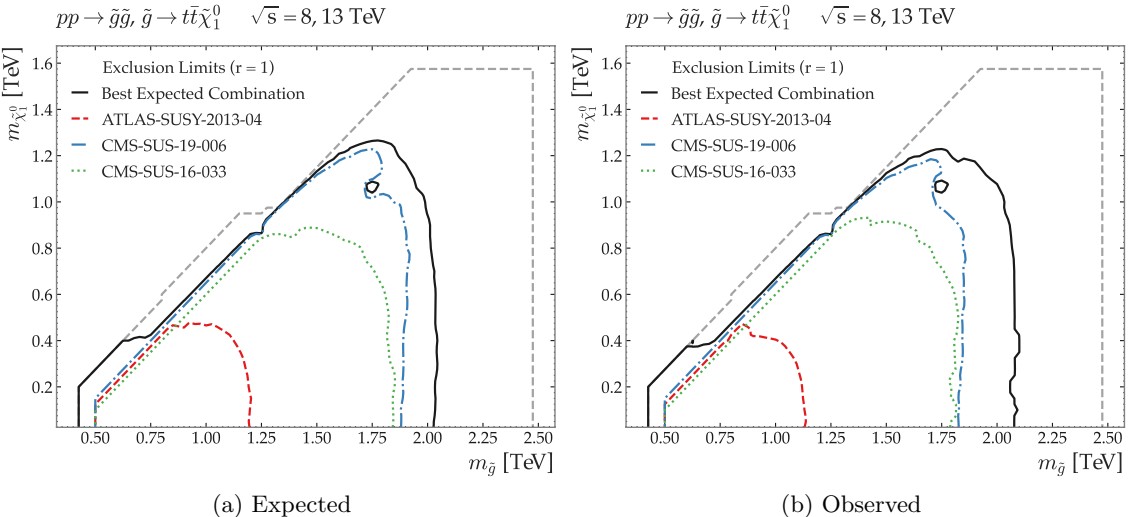

(a) Expected             (b) Observed

Figure 7: Validation plots comparing (a) the expected and (b) observed results of the combination results with 3 single analyses available in the SMODELS results database for the T1tttt topology. The lines show the exclusion limits in terms of $r$-value where $r = 1$ is analogous to the 95% confidence level. The dashed-grey line indicates the boundary of the efficiency maps.

We start with the expected $P(\text{combined}|\text{single})$ result shown in the top-left of Figure 9 (a), which shows how TACO combination changes the $p$-values of model points given their initial $p$-value as obtained from the single best-expected SR. The overall form of the transition pattern is logically consistent with the aim to use SR combinations to *increase* exclusionary ability, thus it is expected that all transitions be located in the upper triangle of the $P(\text{combined}|\text{single})$ matrix. Encouragingly, the transitions are dominated by movements into the excluded $p \in [0, 0.05)$ bin, not just from neighbouring "nearly excluded" single-SR bins, but across the whole spectrum of single-SR $p$-values: this shows that even 40% of the least excluded single-SR points can be excluded when combination of independent SRs is enabled. Some subleading transition structures can be seen within the matrix, showing below-threshold increases in exclusion which can potentially be brought above threshold by availability of more analyses. The expected $P(\text{single}|\text{combined})$ results in Figure 9 (b) are concentrated in the lower triangle of the matrix, as expected, with similar evidence of structures in the transition pattern.

Moving to the observed case in the lower plots of Figure 9, the results become more nuanced as the distribution of transition becomes dilute. The dominant transition into the exclusion bin identified in plot (a) is replicated in the observed case shown in plot (c), as was expected from the histogram results. The "negative transition" of model points in the direction opposite to expectation is caused by over-fluctuations in the observed yields of the SRs used to calculate the combined result. This may be a statistical feature intrinsic to combining multiple SRs, although only present in a small minority of cases ($\approx 5\%$). Looking back to Figure 8 (b) it can be seen that the percentage of points in the exclusion bin jumps from $\approx 35\%$ to over 90% when using the combined SRs. Thus, the negative transitions seen in plot (d) of Figure 9 are only representative of a small fraction of model points.

Figure 10 shows the pMSSM-19 wino-LSP reinterpretation results. When compared to the bino results in Figure 8 we can see a similar overall shift to the exclusion bins, with

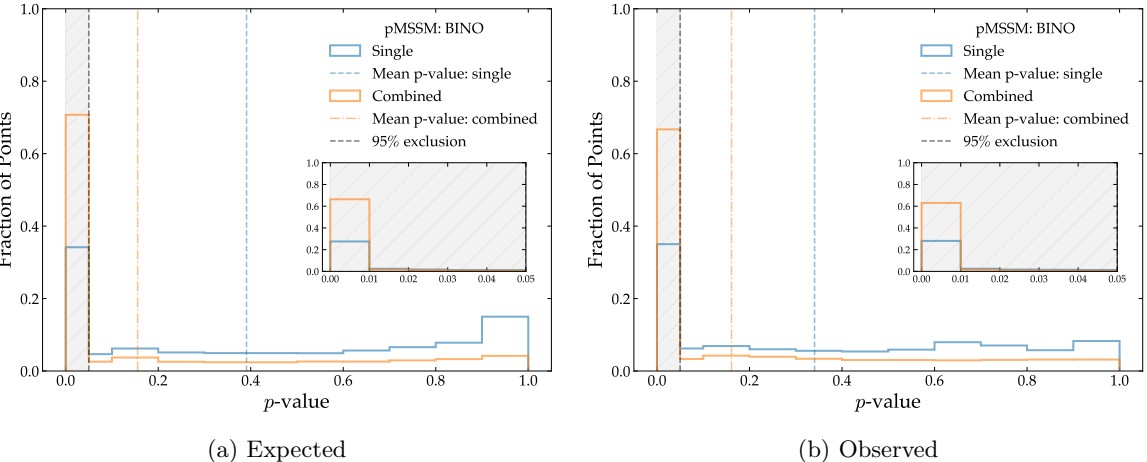

(a) Expected

(b) Observed

Figure 8: Results from the pMSSM-19 bino reinterpretation using the TACO combination method. $p$-values were calculated from a selection of $22\,000$ points taken from the ATLAS pMSSM-19 data set. The blue and orange dashed lines show the mean $p$-values for the single and combined results respectively. The histograms show that in both the (a) expected and (b) observed cases a large fraction of points are moved beyond the 95% exclusion limit by WHDFS SR-combination.

the migration into the 95%-exclusion bin now from approximately 15% (single) to over 50% (combined) in both the expected and observed cases. Notably the wino scenario has a larger fraction of expected single-SR points at $p$-values $> 0.3$, meaning there is a larger population of points at moderate and high-$p$ able to be improved upon by SR combination.

Considering the transition matrices shown in Figure 11, we see similar trends to the bino case. The $P$(combined|single) plots show the clear shift into the exclusion bin identified in the 1D histogram, and the observed plots again contain a degree of negative transitions, although not enough for the overall population of higher-$p$ bins to increase.

The expected transitions into the 95%-excluded $p$-value bin extend less far along the single-SR $p$-value spectrum than in the bino-LSP case, with only 12% of the least-excluded single-SR points (those in the single-SR $p > 0.95$ bin) expected to transition into the combined-SR exclusion bin. In practice, seen in the observed-yield plots, over-fluctuations in SR yields led to greater exclusion than expected for poorly constrained single-SR model points, with nearly 50% of the least excluded being eliminated in combination.

The filament structures in both the expected and observed sets of transition plots are more prominent than in the wino case, allowing identification of their origin. One, the shallower lower line in subfigure (b), can be identified with the single-SR peak structures at $\sim 0.95$ and $\sim 0.65$ for expected and observed respectively; the other is the main trend of migration, showing that the dominant contribution to a given combined $p$-value bin comes from a single-SR $p$-value bin 0.2 units higher. These transition structures again highlight potential for further improvements in model-point exclusion fraction upon availability of more SRs.

As in the simplified-model interpretation of Section 4.1, we can examine the performance of the WHDFS SR-combination algorithm in both the bino- and wino-LSP pMSSM-19 reinterpretations. The distributions of initial SR indices are shown in the upper row of Figure 12, with the same bias toward low starting indices as in Figure 6.

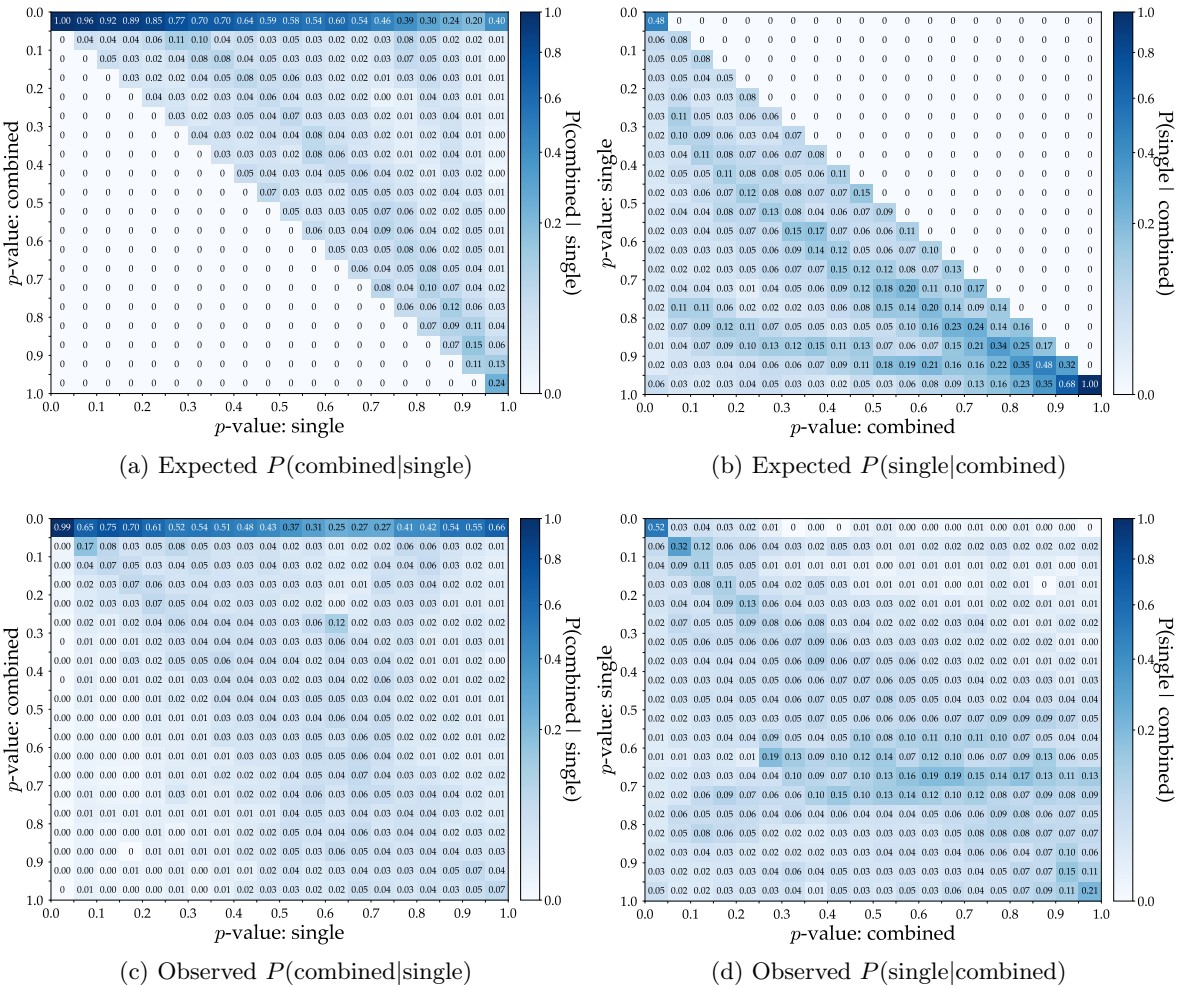

(a) Expected $P(\text{combined}|\text{single})$

(b) Expected $P(\text{single}|\text{combined})$

(c) Observed $P(\text{combined}|\text{single})$

(d) Observed $P(\text{single}|\text{combined})$

Figure 9: Transition matrices showing the pMSSM-19 bino results. The matrices describe the probability of a model point moving from one $p$-value bin to another, by use of the SR-combination scheme rather than the conservative single-best-SR strategy adopted by current recasting tools. The subfigure columns split the transition behaviours by the transition to combined $p$-value distributions given single-SR performance, and the single-SR origins of each combined-SR $p$-value range. The top and bottom subfigure rows show the expected and observed transitions respectively.

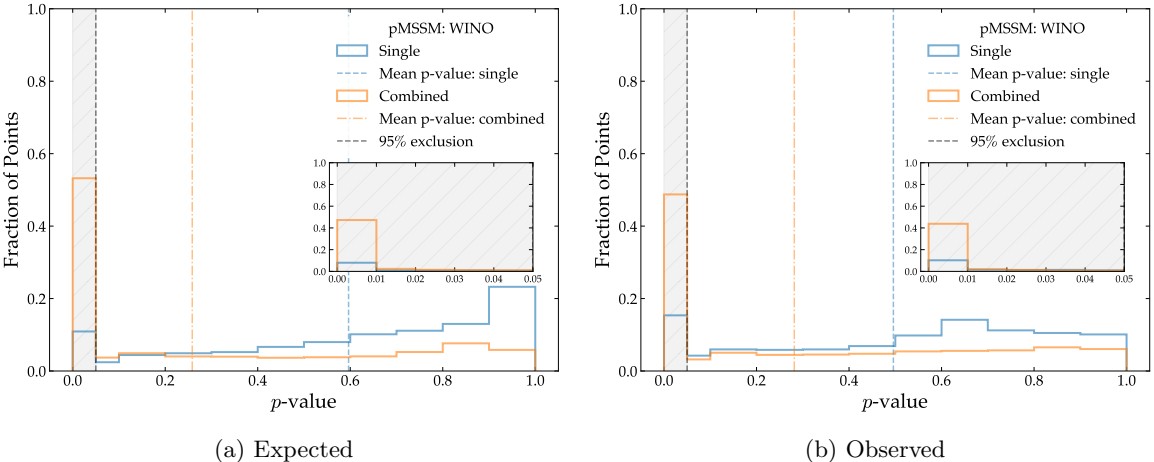

(a) Expected

(b) Observed

Figure 10: Results from the pMSSM-19 wino reinterpretation using the TACO combination method. $p$-values were calculated from a selection of 20 000 points taken from the pMSSM-19 data set. The blue and orange dashed lines show the mean $p$-values for the single and combined results respectively. The histograms show that in both the (a) expected and (b) observed cases a large fraction of points are moved beyond the 95% exclusion limit by WHDFS SR-combination.

The lower two plots of Figure 12 show the distribution of the number of SRs per combination for (c) the bino and (d) the wino reinterpretations. The rapid fall-off of this distribution is a direct consequence of the hereditary condition which reduces the number of available SRs with each iteration of the path-building. Thus, there is a critical point beyond which the cumulative drop-off of available SRs becomes statistically evident in the mean number of SRs. For the pMSSM-19 bino and wino analyses, this critical point occurs around 11 SRs.

## 4.3 $t$-channel dark-matter

As a final illustration of the strength of our approach, we consider in this section one of the $t$-channel dark matter models explored in Refs. [42, 43]. Here, the Standard Model is extended by one fermionic dark-matter candidate $\chi$ and one scalar mediator state $Y$, which interact with the right-handed up-quark. The model's Lagrangian reads

$$\mathcal{L} = \mathcal{L}_{\mathrm{SM}} + \mathcal{L}_{\mathrm{kin}} + \left[ y\big(\chi u_R\big) \, Y^\dagger + \mathrm{H.c.} \right], \qquad (11)$$

where $\mathcal{L}_{\mathrm{SM}}$ is the Standard-Model Lagrangian, $\mathcal{L}_{\mathrm{kin}}$ contains kinetic and mass terms for all new states, and $y$ dictates the strength of the interaction between the mediator, the dark matter and the up-quark. In such a model, the *full* new-physics signal contains three contributions, namely

1. direct dark-matter production in association with one hard jet originating from initial-state radiation $(pp \to \chi\chi j)$;

2. on-shell mediator-pair production followed by mediator decays into dark matter and jets $(pp \to YY^* \to \chi j \chi j)$;

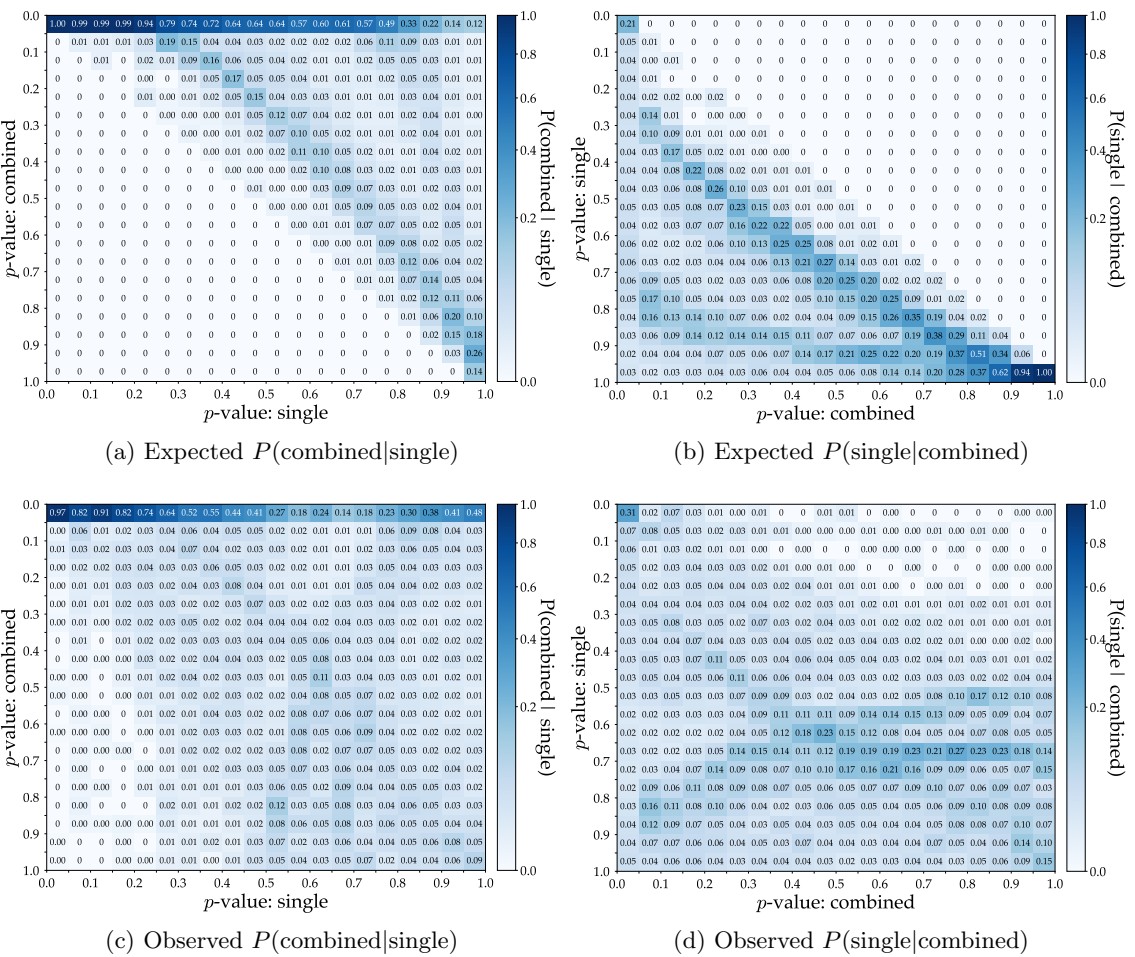

(a) Expected $P(\text{combined}|\text{single})$

(b) Expected $P(\text{single}|\text{combined})$

(c) Observed $P(\text{combined}|\text{single})$

(d) Observed $P(\text{single}|\text{combined})$

Figure 11: Transition matrices showing the pMSSM-19 wino results. The matrices describe the probability of a model point moving from one $p$-value bin to another, by use of the SR-combination scheme rather than the conservative single-best-SR strategy adopted by current recasting tools. The subfigure columns split the transition behaviours by the transition to combined $p$-value distributions given single-SR performance, and the single-SR origins of each combined-SR $p$-value range. The top and bottom subfigure rows show the expected and observed transitions respectively.

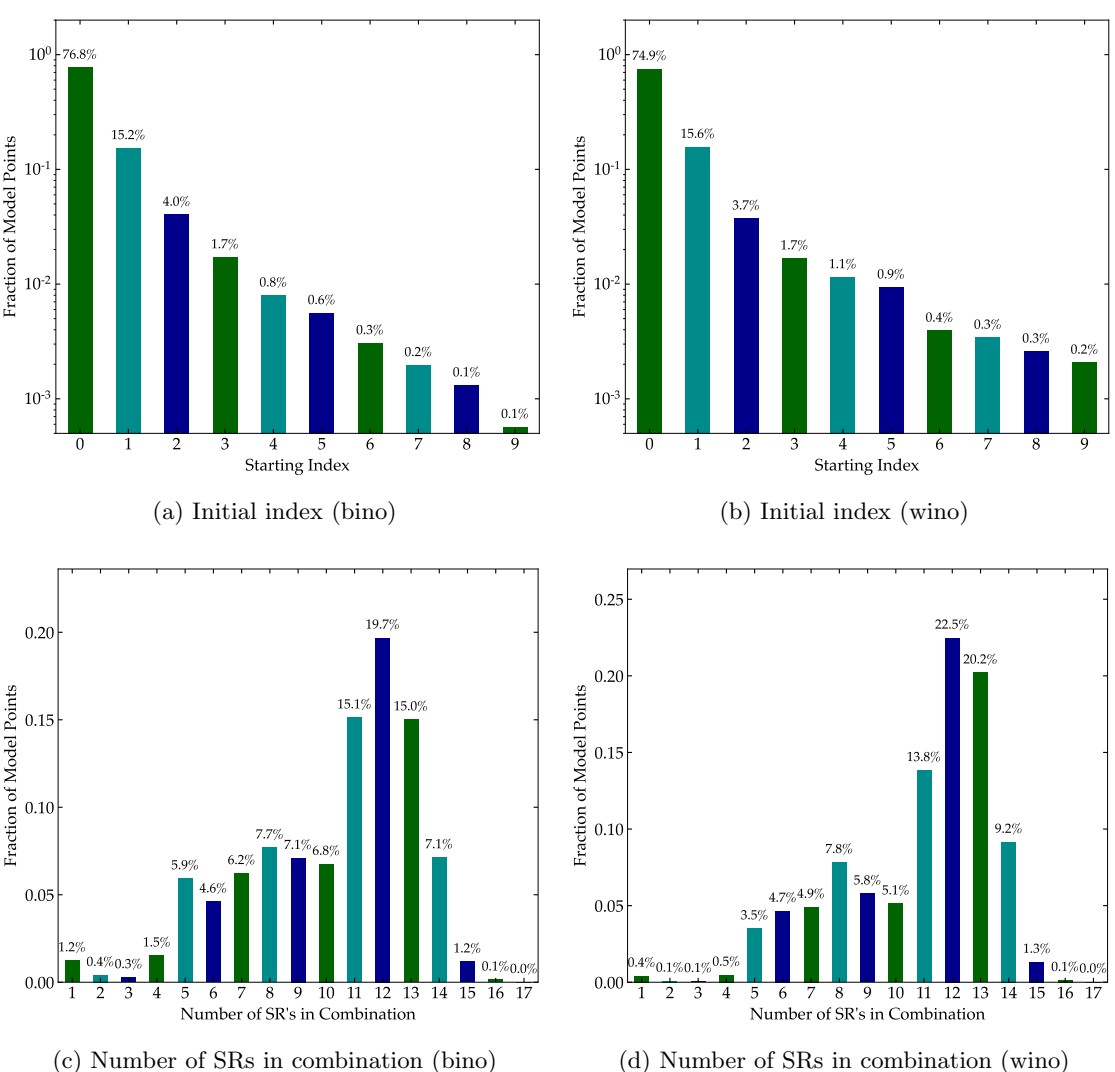

(a) Initial index (bino)

(b) Initial index (wino)

(c) Number of SRs in combination (bino)

(d) Number of SRs in combination (wino)

Figure 12: Fractional distributions of starting-SR (a, b) and number of SRs (c, d) in each combination. Data is taken from the optimum SR-combination found for each model-point for both the pMSSM-19 bino and wino reinterpretations. As mentioned in Section 3 the combinations are constructed from an differently ordered set of SRs for each point in the model space, so the identity of the zeroth SR is free to change from point to point.

3. and the associated production of a mediator (that then decays into a $\chi j$ system) and a dark-matter state ($pp \to \chi Y \to \chi(\chi j)$).

Such a signal can be searched for through analyses targeting the production of multiple jets in association with missing transverse-energy, each component of the signal yielding a different jet-multiplicity spectrum and different jet properties. We therefore focus on the reinterpretation of the results of the ATLAS-SUSY-2015-06 [13], ATLAS-SUSY-2016-07 [14], CMS-SUS-16-033 [22] and CMS-SUS-19-006 [26] analyses to investigate which mediator and dark-matter mass configurations are allowed by data, for a new-physics coupling set to $y = 1$. All analyses considered are integrated in the MADANALYSIS 5 Public Analysis Database [44], the recast codes and their detailed validation notes being available from Refs. [45–49] and on the database webpage.[2]

To estimate the individual exclusion limits originating from each analysis, we used MAD-GRAPH5_aMC@NLO v2.6.5 [29] to generate hard-scattering events at leading-order (LO) accuracy. We grouped the three different contributions to the signal in two sets according to the parton-level jet multiplicity. A first matrix-element describes the production of a pair of dark-matter states with a single hard jet ($pp \to \chi\chi j$), and a second one concerns mediator pair-production and decay ($pp \to YY^* \to \chi j \chi j$). The associated production of a dark-matter particle with a mediator is hence included in the first subprocess, as it yields the same final-state ($pp \to \chi Y \to \chi(\chi j)$ with the intermediate mediator $Y$ being on-shell). These two matrix-elements were convolved with the NNPDF 2.3 LO [30] set of parton distribution functions, and we generated 200 000 signal events per model-point to limit statistical uncertainties. Hadronisation and parton showering were handled with PYTHIA 8 v8.240 [50] and the simulation of the CMS and ATLAS detector responses was approximated with DELPHES 3 [33], using the custom detector-parameterisation provided with each recast code.

The results are displayed in the ($m_Y, m_\chi$) plane in Figure 13, for a mediator mass $m_Y \in [0.5, 1.8]$ TeV and a dark-matter mass $m_\chi \in [0.1, 1]$ TeV. We separately show exclusions extracted from single-jet events only ($pp \to \chi\chi j$, upper left panel) and from dijet events only ($pp \to YY^* \to \chi j \chi j$, upper right panel), as well as the combined limits derived by considering the *full* new-physics signal (lower panel). In these three figures, the dashed lines represent the individual limits obtained by the reinterpretation of the results of the ATLAS-SUSY-2015-06 (purple), ATLAS-SUSY-2016-07 (green), CMS-SUS-16-033 (blue) and CMS-SUS-19-006 (orange) analyses, which respectively probe integrated luminosities of $3.2\,\mathrm{fb}^{-1}$, $36.1\,\mathrm{fb}^{-1}$, $35.9\,\mathrm{fb}^{-1}$, and $137\,\mathrm{fb}^{-1}$. These limits were obtained by conservatively considering the signal region of a given analysis giving rise to the best expected exclusion, for a given benchmark point.

Unsurprisingly, the analysis making use of the largest amount of data, CMS-SUS-19-006, leads to the strongest individual exclusion. Considering only the "single-jet" component of the signal (Figure 13, upper left panel), mediator masses up to $1.5\,\mathrm{TeV}$ are excluded by the CMS-SUS-19-006 analysis for small dark-matter masses $m_\chi$. By comparison, the ATLAS-SUSY-2016-07 and CMS-SUS-16-033 analyses which analysed only one third of the Run 2 integrated luminosity, are only sensitive to mediator masses smaller than $900\,\mathrm{GeV}$ to $1000\,\mathrm{GeV}$ for the same $m_\chi$ assumptions. Similarly, scenarios with more compressed spectra, which are intrinsically harder to probe as they correspond to the production of softer final-state objects, see better coverage from the most recent analysis than from the two partial Run 2 analyses.

---

[2]See http://madanalysis.irmp.ucl.ac.be/wiki/PublicAnalysisDatabase.

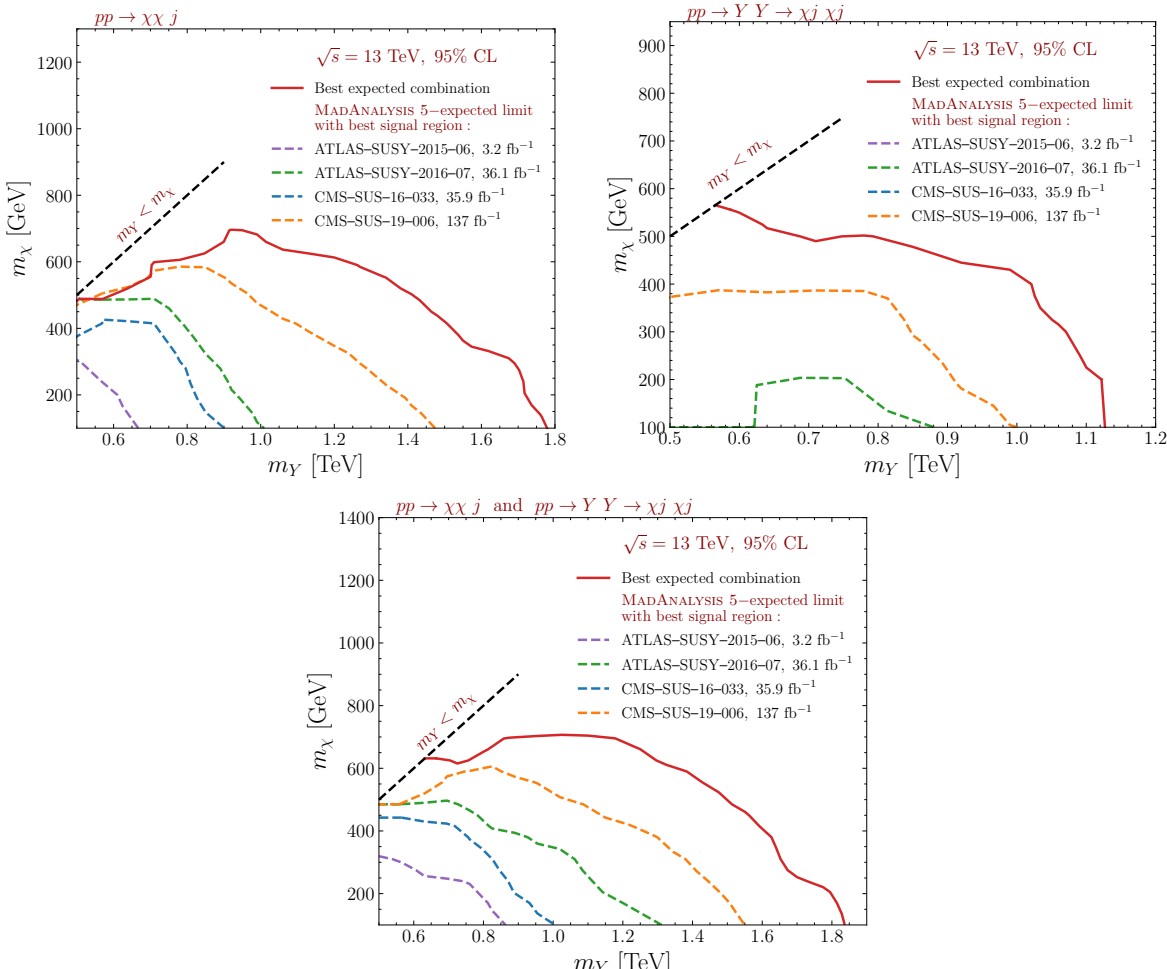

Figure 13: 95% confidence level exclusions for the studied $t$-channel simplified model of dark matter. We explore separately the two components of the new-physics signal respectively arising from the processes $pp \to \chi\chi j$ (upper left) and $pp \to YY^* \to \chi j \chi j$ (upper right), as well as their sum (lower panel). We display exclusion limits originating from the individual recast analyses, *i.e.* for the ATLAS-SUSY-2015-06 (dashed purple line), ATLAS-SUSY-2016-07 (dashed green line), CMS-SUS-16-033 (dashed blue line) and CMS-SUS-19-006 (dashed orange line) analyses, as well as those derived from their combination through the method proposed in this paper (solid red line).

By contrast to all of these, the early- Run 2 ATLAS-SUSY-2015-06 analysis barely reaches exclusion for new-physics masses at 500 GeV.

Similar conclusions hold for the "dijet" component of the signal (Figure 13, upper right panel). The most recent CMS-SUS-19-006 analysis is sensitive to both larger new-physics masses and more compressed spectra than the partial Run 2 analyses — the latter are in this case barely sensitive to any signal. The sensitivity is found to be significantly milder than for the single-jet signal component, due to the phase-space suppression associated with the production of two heavy mediators. Consequently, it turns out that for the range of masses to which Run 2 of the LHC is sensitive, the limits obtained for the full new-physics scenario (Figure 13, lower panel) are almost identical to these obtained in the single-jet scenario. For light dark matter, we observe an improvement of about 100 GeV; this demonstrates that even when it yields mild effects, use of the full new-physics signal is always better than an approximate modelling that does not include all relevant subprocesses.

Figure 13 also shows the impact of combining the four analyses, first for the individual components of the new-physics signal (upper panel), and next after combining them (lower panel). This was enabled by determining and using the overlaps between the various analyses' signal regions cf. Sections 2 and 3. Despite the fact that the analyses targeted similar topologies (multijet and missing energy), some of their signal regions proved uncorrelated enough for combination to be performed. This ilustrates the advantage of an objective and quantified measure of acceptance overlap in place of an informal guesstimate of orthogonality. The allowed level of combination allows for an increase in parameter-space coverage, as displayed by the solid red contours in Figure 13. We observe a substantial improvement of the limits through WHDFS SR-combination, both for split-mass spectra (light dark-matter and heavy mediator) and compressed spectra. For $m_\chi \approx 100$ GeV, mediator masses ranging up to 1.9 TeV are reachable, whereas scenarios with a dark-matter mass $m_\chi < 600$ GeV get excluded for $m_Y < 1.2$ TeV.

# 5 Conclusions and outlook

In this paper we have argued that maximising the BSM-search power of the LHC and other colliders, in the light of Run 2's significant list of null-result direct searches, necessitates combination of analyses for sensitivity to more subtle dispersed-signal models than have so far been considered.

But combination cannot be performed naively, due to overlaps in analyses' event acceptance. Short of many-year, top-down coordination within experimental collaborations to forbid phase-space overlaps (with implicit prioritisation of some analyses over others) or public lists of which collider-event numbers entered which signal regions across all published analyses, a *post hoc* method is needed to estimate the extent of such overlaps. In this paper we have presented the TACO method for this estimation, using the SMODELS and MADANALYSIS 5 analysis databases to guide simulated-event population of all recastable signal regions. A new augmentation to the MADANALYSIS 5 analysis machinery enables the overlap coefficients between pairs of signal regions to then be computed via Poisson bootstrapping.

We have also shown how this information can be used in a scalable way to obtain the expected optimally model-excluding, non-overlapping subset of signal regions for a given BSM model or model-point. The combinatorically hard problem of evaluating all allowed subsets

of $\mathcal{O}(400)$ signal regions (at present, and guaranteed to grow) is mapped to construction of directed acyclic graphs representing SR combinations. This construction is made tractable and even rapid by use of a binarised form of the SR-overlap matrix to efficiently exclude sequences of partially overlapping graphs, and by ordering SRs in the graph construction according to their expected log likelihood-ratios. The expected best-sensitivity combination can then be efficiently identified by a weighted hereditary depth-first search (WHDFS) algorithm, in direct analogy to a weighted longest-path problem.

Code for both the overlap estimation and SR-combination aspects of this paper is publicly available from the https://gitlab.com/t-a-c-o/taco_code online repository.

Computation of such SR-subsets using the TACO WHDFS algorithm is a practical alternative to direct use of the overlap matrix to compute a correlated $\chi^2$ or other measure across all signal regions, given the latter approach's high risk of numerical instabilities in covariance inversion, and the huge computational cost of simultaneously likelihood-profiling over a full set of SRs. This graph-based approach hence has potential not just as a route to composite likelihoods in reinterpretation, but also as a dimension-reduction technique for visualising and interpreting how dominant categories of analyses and signal-regions evolve through model spaces.

We have tested the TACO overlap-estimation and optimal-subset computations against several BSM models of increasing complexity: a SUSY simplified model, ATLAS' 8 TeV scan of pMSSM-19 points re-evaluated on 13 TeV measurements, and a $t$-channel dark-matter model. In all cases we see the algorithmic combination of SRs providing a significant increase in experimental limit-setting reach, typically $\mathcal{O}(100)$ GeV in the mass parameters of both the simple and complex BSM models considered with currently available reinterpretation analyses. The study of transition matrices for the SR combination indicates that the gains obtained are not just marginal — moving already near-exclusion model points over the line — but holistic, with evidence that combination of up to ten relatively weak signal regions can create a complementary strong limit.

This method hence demonstrates that *post hoc* combination of BSM direct-search data is possible and can be made computationally efficient, and that pessimistic use of at most one signal-region from each event topology is no longer necessary. As an efficient and empirical computational method, TACO is scalable to hundreds of potentially overlapping analyses, beyond the capacity of manual and fallible assessment of uncorrelated analysis sets. It is, of course, imperfect. The assumption of effective orthogonality of finitely overlapping $\rho_{ij} < T$ signal regions is key to efficient computation of SR subsets, but a hybrid of subset-identification with correlated LLR evaluation on the reduced set is easily appended to the procedure described here. Systematic uncertainties are also missing from the current treatment, but — at least on the subset of uncertainties that can be evaluated by event reweighting — this is again not an intractable problem. We hope that this method and toolkit will prove a useful target for how collider BSM-combinations are designed and performed in the coming years, with submission of analysis routines to the key reinterpretation-analysis frameworks and provision of event-bootstrapping machinery beyond MADANALYSIS 5.

# Acknowledgements

AB and JY thank the Royal Society for studentship funding under URF Enhancement Grant RGF\EA\180252, and STFC for support under Consolidated Grant ST/S000887/1. The research of HRG is funded by the Italian PRIN grant 20172LNEEZ. The authors thank the Les Houches Physics at the Terascale series for its 2019 edition during which this work originated, and we look forward to the return of collider-physics to the academically productive slopes of the Chamonix valley. Our further thanks to Sabine Kraml and others at LPSC Grenoble for generously bringing the authors together to invigorate this work, with support from the IN2P3 project "Théorie – BSMGA".

# A   BSM-search overlap matrices

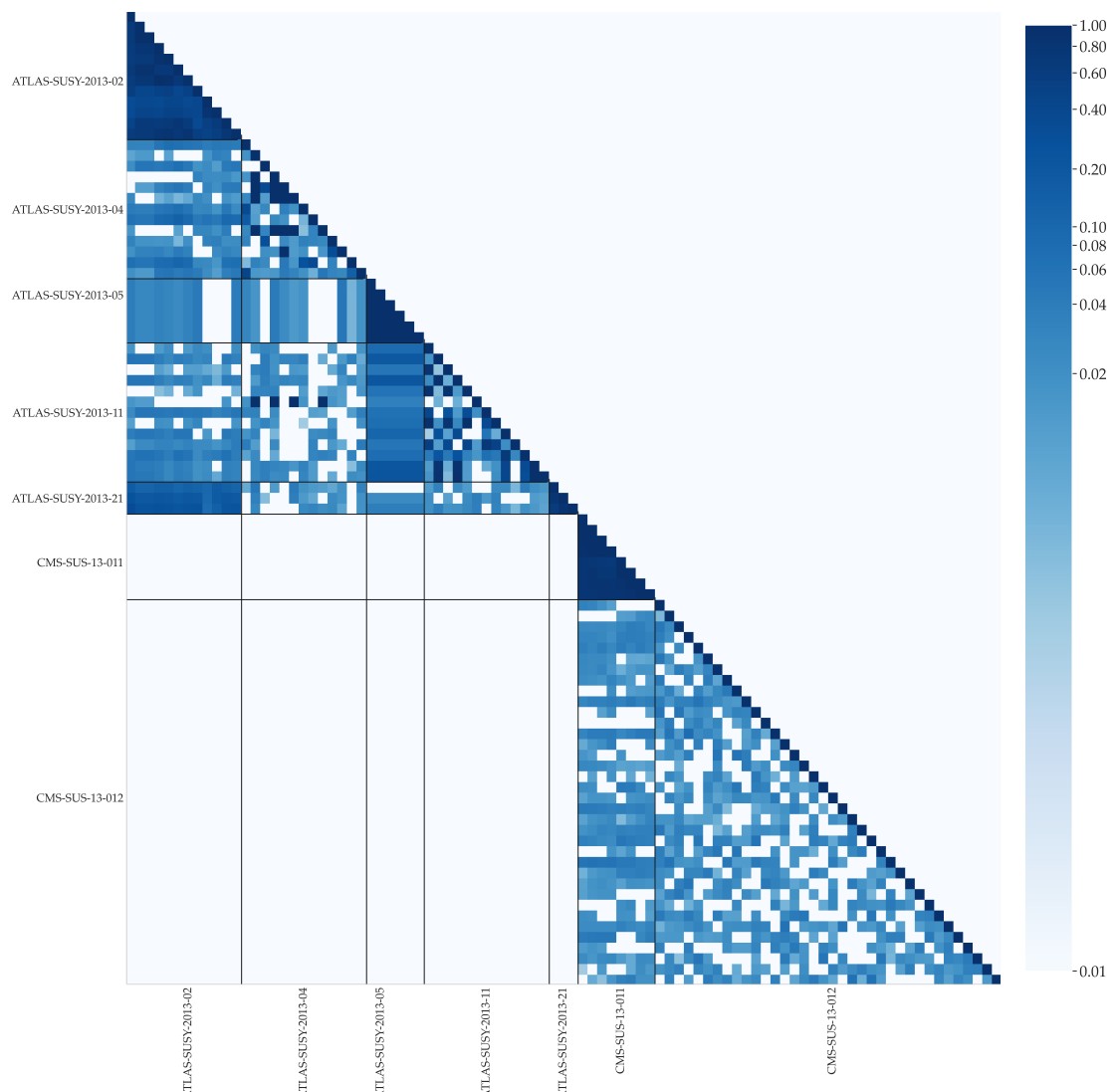

Figure 14: The overlap matrix $\rho_{ij}$ obtained from the TACO sampling procedure between all LHC BSM searches at 8 TeV commonly implemented in SMODELS and MADANALYSIS 5. Non-overlap between ATLAS and CMS analyses is manually imposed, as the same proton-collisions could not be accepted by analyses from both experiments regardless of the MC overlaps, and similarly overlaps between 8 and 13 TeV analyses must be zero regardless of final-state acceptances. The set of SR pairs considered sufficiently independent in the analyses of Sections 3 and 4, with $T < 0.01$, are shown in white.

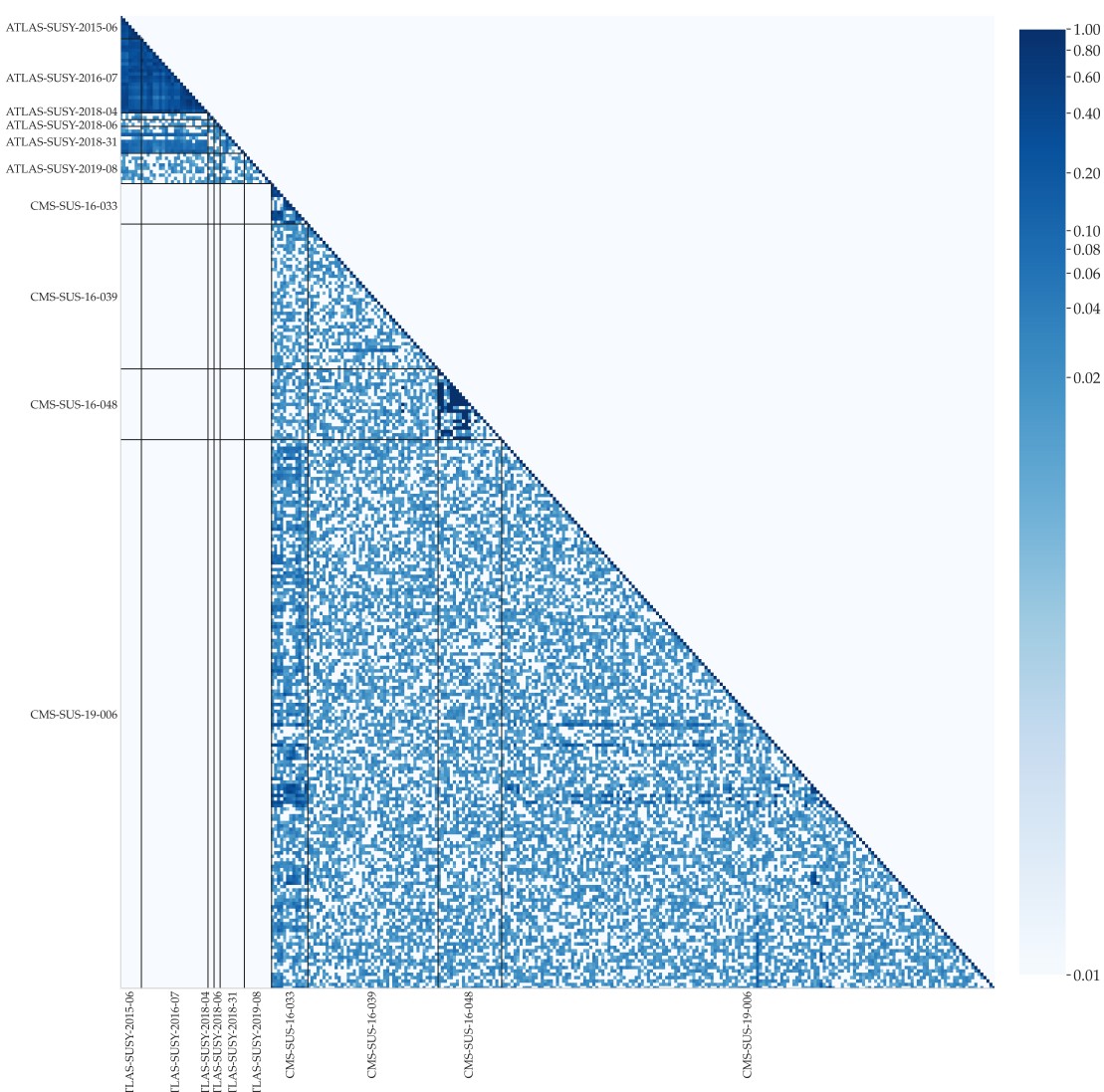

Figure 15: The overlap matrix $\rho_{ij}$ obtained from the TACO sampling procedure between all LHC BSM searches at 13 TeV commonly implemented in SMODELS and MADANALYSIS 5. Non-overlap between ATLAS and CMS analyses is manually imposed, as the same proton-collisions could not be accepted by analyses from both experiments regardless of the MC overlaps, and similarly overlaps between 8 and 13 TeV analyses must be zero regardless of final-state acceptances. The set of SR pairs considered sufficiently independent in the analyses of Sections 3 and 4, with $T < 0.01$, are shown in white.

# B HDFS algorithm

The pseudocode shown in Algorithms 1 and 2 are written in a Pythonic syntax as the code makes use of the generator – donoted by the term *Gen()* – functionality which allows for efficient iteration ordering. Aspects of the code are heavily influenced by the "all simple paths" method from the Python NETWORKX package [39].

---

**Algorithm 1** Hereditary Depth-First Search (HDFS)

---

**Require:** source = i
  Target = n
  Cutoff = $n-1$
  Visited = $[i]$
  Stack = $[Gen(A_i)]$
  $\mathcal{S} = [A_i]$
  **while** Stack is not empty **do**
    Children = last element of stack
    c = Next element in Children **or** None **if** Empty
    **if** c is None **then**
      Drop last element of Stack
      Drop last element of $\mathcal{S}$
      Drop last element of Visited
    **else if** length(Visited) < cutoff **then**
      **if** c = Target **then**
        **Yield**: Visited + [c]
      **end if**
      Visited += [c]
      **if** target not in Visited **then**
        $\mathcal{S}_c = A_c \cap S_{c-1}$
        Stack += [Gen(j: **for** index in $A_c$ **if** index $\in \mathcal{S}_c$)]
      **else if**   **then**
        Drop last element of Visited
      **end if**
    **else if** length(Visited) = cutoff **then**
      Drop last element of Stack
      Drop last element of $\mathcal{S}$
      Drop last element of Visited
      **Yield**: Visited + [Target]
    **end if**
  **end while**

---

# C WHDFS algorithm

The pseudocode shown in Algorithm 2 is a modification of Algorithm 1. WHDFS uses the edge weights to calculate an upper limit of total weight available at each step in the path. This modification eliminated the need to explore all allowed paths instead limiting the combinations to those that have the greatest potential.

---

**Algorithm 2** Weighted Hereditary Depth-First Search (WHDFS)

---

**Require:** source = i
**Require:** maximum weight
  Best Path = []
  Target = n
  Cutoff = $n-1$
  Visited = $[i]$
  Stack = $[Gen(A_i)]$
  $\mathcal{S} = [A_i]$
  **while** Stack is not empty **do**
      Children = last element of stack
      c = Next element in Children **or** None **if** Empty
      **if** c is None **then**
         Drop last element of Stack
         Drop last element of $\mathcal{S}$
         Drop last element of Visited
      **else if** length(Visited) < cutoff **then**
         **if** Target in Visited **then**
            continue
         **end if**
         $\mathcal{S}_c = A_c \cap S_{c-1}$
         Visited += [c]
         Current Weight = weight function (Visited)
         Available Weight = weight function $(\mathcal{S}_c)$
         **if** c = Target & Current Weight > Max weight **then**
            Max weight = Current Weight
            Best Path = Visited
         **end if**
         **if** (Current Weight + Remaining Weight) > Max weight **then**
            Stack += [Gen(j: **for** index in $A_c$ **if** index $\in \mathcal{S}_c$)]
         **else if**  **then**
            Drop last element of Visited
         **end if**
      **else if** length(Visited) = cutoff **then**
         Drop last element of Stack
         Drop last element of $\mathcal{S}$
         Drop last element of Visited
      **end if**
  **end while**
  **return** Best Path

---

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
