# Peer review of "Strength in numbers: optimal and scalable combination of LHC new-physics searches"

_SciPost Physics, doi:SciPost Phys. 14, 077 (2023)_

## Round 1 · Referee Report · Andrew Fowlie (Referee 1) · 2022-9-27

Strengths

  1. Solves a genuine problem in combining overlapping LHC searches
  2. Realistic examples and public code integrated into existing software ecosystem
  3. Benefits clearly demonstrated; it leads to improved constraints in models of new physics
  4. Potentially a significant development that changes the way we use LHC data

Weaknesses

  1. Presentation unclear in a few places
  2. Unclear notions of power throughout

Report

I have read the paper 'Strength in numbers: optimal and scalable combination of LHC new-physics searches'. This paper addresses a real problem in collider phenomenology: how to deal with overlapping searches. They work in a frequentist framework, such that the problem may be stated as how to pick a set of disjoint searches in such a way that the the power is maximised.

The paper proposes an algorithm for identifying the most powerful set of disjoint searches, and demonstrates the increase in power as well as changes in observed limits in a few new physics scenarios. The realistic examples and the fact that the work is accompanied by a public code gives me confidence that this development could really help us get the most out of LHC data.

The paper should undoubtedly be published. However, I think the presentation and clarity could be improved in a few places. I have quite a few comments, mainly because the paper was interesting and important. None of my comments below challenge the novelty, significance or correctness of the methods or results, though I do think the authors should try to address them.

Requested changes

1.

a) I thought that the introduction should emphasise that this is partly a missing data problem, and better explain the notion of statistical power that they will be using. 'Power' seems to be used informally in a few ways. I wasn't sure whether they meant

power = Probability we reject NP scenario when SM only is true

or

power = Probability we reject SM when NP scenario is true

given that I read 'exclusion power' perhaps I guess the first one. Some mention of this choice would be helpful and why focus on exclusion rather than discovery. They may in many cases be quite similar.

2.1

a) I found the first paragraph of 2.1 difficult to follow and suggest it is rewritten. My understanding is that:

  • the results of the searches are expressed in terms of simplified models
  • the overlap between searches depends on the choice of simplified model parameters
  • we want to find searches that are disjoint for every possible choices of simplified model parameters

and for this reason, we try to densely sample the simplified model parameter space and compute a sort of expected overlap between searches. We want to find sets of searches for the expected overlap is zero. If my understanding is correct, some more explanations are necessary, such as: from what density are the parameters sampled? Why do we desire searches that are disjoint for every possible parameter choice in the simplified models? as opposed to searches that are disjoint for the simplified model parameters of a specific scenario under consideration?

b) The algorithm depends on a number of fixed parameters. First, if k > 100, it is judged that enough events were generated. If k <= 100, even generation continues. We compute a 95% upper limit on p, denoted by f. If f < 1%, 'we have accumulated enough statistics to safely infer the potential absence of an overlap, and we confidently proceed to the bootstrapping procedure.'

i) Presumably, it should say something more like, we have accumulated enough statistics for safely infer that any overlap can be neglected. We cannot conclude that the overlap is zero. Similarly, regarding the closing remark,

It will guarantee that enough statistics is available to robustly and reliably determine both the existence as well as lack of an overlap between a given pair of signal regions.

The procedure may determine whether overlap is negligible (according to some criteria) but not the absence of an overlap.

ii) Why can f < 1% be neglected? How was this threshold chosen?

iii) Initially, we allow k > 100 events, an unbiased estimator of f gives 101 / 10 000 ~ 0.1%, with a 95% upper limit of around 0.11%. This doesn't seem to be consistent with neglecting f < 1%. I suppose these are f < 1% contributions so whether they are neglected entirely or included is not important.

iv) I don't believe the confidence interval correctly accounts for the optional stopping in this procedure, though this isn't particularly important

2.2

a) This section describes a Poisson bootstrapping procure to estimate a correlation matrix. To be honest, I didn't understand why this procedure was done. My limited knowledge of bootstrap is that it would typically be used to estimate the bias, standard error and more generally distribution of a test-statistic or estimator. I don't see why we should consider a mean of the bootstrap realisations of an estimator rather than just the estimator. Typically, the mean of the bootstrap of realisations of the estimator is just the estimator.

Let me go through an example to explain my point of view. Let's suppose we look at two regions A and B. Denote the unique events in A by U_A, and in B by U_B, and the events in both by O_AB, such that region A contains a total of U_A + O_AB events. A simple estimator of the covariance would be

O_AB / sqrt((U_A + O_AB) (U_B + O_AB))

Now consider the Poisson bootstrap. Our counts U_A, U_B and O_AB are replaced by draws from a Poisson with means U_A, U_B and O_AB. This is equivalent to replicating each event r times where r ~ Poisson(1). Let me use lowercase letters to denote the draws, such that u_a ~ Poisson(U_A) etc.

Consider covariance between the counts in A and B, you can quickly compute

< (u_a + o_ab) (u_b + o_ab) > - < (u_a + o_ab) > < (u_b + o_ab)> = O_AB

The variance of counts in A is just U_A + O_AB and similarly for B. And so the correlation would be

rho = O_AB / sqrt((U_A + O_AB) (U_B + O_AB))

In other words, the expectation of the bootstrap is just the simple estimator, and in fact it would return it exactly in the limit N_B \to \infty. So why did we do the bootstrap?

b) How was the threshold T chosen? Can we say anything about the error that is introduced by allowing 0 < \rho < T? Why would I take a nonzero T? Presumably there is some trade-off here between fidelity (the regions are actually overlapping) and power (allowing tiny overlaps is presumably a reasonable approximation and gains power). Some comments would be in order.

c) It may be convenient to define E = \rho \le T, as then it works when we take T = 0, which may be desirable.

a) In eq. 5, why omit r = 1? Why can't a solution be a single region?

b) Above eq. 7, 'We can now expand on the previous definition of K'. The symbol K was used previously, but hardly defined. As a result, I found this discussion hard to follow.

c) The 'expected Poisson likelihood-ratios between the signal-model under test and the background expectation' is used a proxy for power. As already mentioned, the paper would benefit from a clearer definition of what the authors mean by power.

i) The 'expected Poisson ...' under what model? Background only or new physics + background? How was this expectation computed numerically?

ii) The connection between this proxy and the power should be commented on. Whilst it is clear that log likelihood between disjoint searches is additive, it is not so clear how power would 'add' or what the authors have in mind when searches are combined.

I guess when combining searches, the authors are thinking about a single test based on a log likelihood ratio for two simple hypotheses. As the searches are independent, the log likelihoods just add. Under what conditions is maximising the sum of expected log likelihood equivalent to maximising power for fixed type 1 error rate? Possibly Wald approximation to this log likelihood ratio is relevant (see e.g., Sec. 3.8 of https://arxiv.org/pdf/1007.1727.pdf), though the details weren't clear to me. The Wald approximation involves two unknowns: the expected log likelihood ratio under the signal + background model, and under the background only model (Eq. 73 with mu = 0 and mu = 1; NB that \sigma Eq. 73 is a function of mu). As the procedure here only involves one of them, it wasn't even clear to me that it maximised power under the Wald approximation.

4.

a) What is the grey dashed line in Fig. 5?

b) I found the grey shading in Fig. 8 made the figure hard to read.

c) It is probably worth saying the distributions in Fig. 8 depend on the way in which the model parameters were sampled and specifying how it was done

d) 'The expected transitions into the 95%-excluded p-value bin are less uniform than in the bino-LSP case, with only 12% of the least-excluded single-SR points expected to transition into the combined-SR exclusion bin' - I couldn't see how Fig. 10 showed this; should Fig. 11 be mentioned already here?

e) I wasn't sure what 'least-excluded single-SR points' meant. Does it mean points in the p-value bin p > 0.95 with single SR?

f) 'This highlights how analysis combination makes limit-setting more robust against fluctuations in single analyses.' I didn't understand this conclusion. What is meant by robust here? The limits are robust against fluctuations in the sense that the type-1 error rate is controlled.

g) 'The asymmetry of the distribution ...' I didn't fully grasp this: asymmetry about the mode? Why would we expect symmetry about the mode in the absence of the hereditary condition? I.e., why is it the hereditary condition that causes this asymmetry?

h) The results in Fig. 13 are impressive! The final paragraph of this subsection that describes them beginning 'In a second step ...' seems quite short and unbalanced compared to the detail given in the other results sections.

  • validity: high
  • significance: high
  • originality: high
  • clarity: good
  • formatting: excellent
  • grammar: excellent

Author:  Humberto Reyes-González  on 2022-11-23  [id 3062]

(in reply to Report 1 by Andrew Fowlie on 2022-09-27)

Our thanks for a comprehensive and detailed read through our paper, which has highlighted a few technical bugs in the presentation and places where more explicit exposition was necessary. We respond to the individual points below, and have uploaded a new version to the arXiv with the issues addressed.

Requested changes 1.

a) I thought that the introduction should emphasise that this is partly a missing data problem, and better explain the notion of statistical power that they will be using. 'Power' seems to be used informally in a few ways. I wasn't sure whether they meant

power = Probability we reject NP scenario when SM only is true

or

power = Probability we reject SM when NP scenario is true

given that I read 'exclusion power' perhaps I guess the first one. Some mention of this choice would be helpful and why focus on exclusion rather than discovery. They may in many cases be quite similar.

We now explain in the introduction that we mean the ability to exclude (for a fixed CL) a BSM model point when SM-only is true, i.e. we minimise the Type 2 error rate for the "exclusion" framing of the BSM model as the null hypothesis. This framing is chosen based on the input data consisting of a set of individual exclusions with no significant evidence of a positive signal: the aim is to combine those sets of exclusions optimally. In practice, of course, the two framings are correlated: an optimal strategy for exclusion power will likely also be a good (but perhaps not optimal) strategy for discovery.

For the examples in this paper, power corresponds to maximising L_b / L_sb under the assumption of SM-like observations because in the Asimov-dataset approach, through Wald's approximation, the expected significance is monotonic to the LLR (typically Z ~ sqrt(q) with q the -2 LLR statistic or a close relative). Maximising the LLR hence also maximises the significance. This is now further clarified in Section 3.

2.1

a) I found the first paragraph of 2.1 difficult to follow and suggest it is rewritten. My understanding is that:

  • the results of the searches are expressed in terms of simplified models
  • the overlap between searches depends on the choice of simplified model parameters
  • we want to find searches that are disjoint for every possible choices of simplified model parameters

and for this reason, we try to densely sample the simplified model parameter space and compute a sort of expected overlap between searches. We want to find sets of searches for the expected overlap is zero. If my understanding is correct, some more explanations are necessary, such as: from what density are the parameters sampled? Why do we desire searches that are disjoint for every possible parameter choice in the simplified models? as opposed to searches that are disjoint for the simplified model parameters of a specific scenario under consideration?

b) The algorithm depends on a number of fixed parameters. First, if k > 100, it is judged that enough events were generated. If k <= 100, even generation continues. We compute a 95% upper limit on p, denoted by f. If f < 1%, 'we have accumulated enough statistics to safely infer the potential absence of an overlap, and we confidently proceed to the bootstrapping procedure.'

i) Presumably, it should say something more like, we have accumulated enough statistics for safely infer that any overlap can be neglected. We cannot conclude that the overlap is zero. Similarly, regarding the closing remark,

It will guarantee that enough statistics is available to robustly and reliably determine both the existence as well as lack of an overlap between a given pair of signal regions.

The procedure may determine whether overlap is negligible (according to some criteria) but not the absence of an overlap.

ii) Why can f < 1% be neglected? How was this threshold chosen?

iii) Initially, we allow k > 100 events, an unbiased estimator of f gives 101 / 10 000 ~ 0.1%, with a 95% upper limit of around 0.11%. This doesn't seem to be consistent with neglecting f < 1%. I suppose these are f < 1% contributions so whether they are neglected entirely or included is not important.

iv) I don't believe the confidence interval correctly accounts for the optional stopping in this procedure, though this isn't particularly important

Done, we have rephrased the text. The bound on f was arbitrary chosen, if f < 1% we assume that we have enough statistics to determine a 'negligible' overlap.

2.2

a) This section describes a Poisson bootstrapping procure to estimate a correlation matrix. To be honest, I didn't understand why this procedure was done. My limited knowledge of bootstrap is that it would typically be used to estimate the bias, standard error and more generally distribution of a test-statistic or estimator. I don't see why we should consider a mean of the bootstrap realisations of an estimator rather than just the estimator. Typically, the mean of the bootstrap of realisations of the estimator is just the estimator.

[...]

In other words, the expectation of the bootstrap is just the simple estimator, and in fact it would return it exactly in the limit N_B \to \infty. So why did we do the bootstrap?

Thank you for the observation! This is a hangover from earlier implementations where the aim was to use bootstrapping to obtain correlation estimates from histograms or summed SR yields only. In that case, the "bootstrap axis" is needed to estimate correlations through the effective event-weight variations. But as our implementation uses an ntuple of per-event SR acceptances, the "event axis" is already available over which to calculate the correlation matrix. However, to not misrepresent the procedure used to obtain the results in this paper, and as the bootstrap approach remains a useful and better-scaling alternative strategy, we now describe both methods and comment on the distinction between them, in the updated text.

b) How was the threshold T chosen? Can we say anything about the error that is introduced by allowing 0 < \rho < T? Why would I take a nonzero T? Presumably there is some trade-off here between fidelity (the regions are actually overlapping) and power (allowing tiny overlaps is presumably a reasonable approximation and gains power). Some comments would be in order.

T was chosen arbitrarily, as perfect decorrelation indeed reduces the number of combinable SRs very pessimistically. Allowing a small degree of finite overlap is exactly a trade-off as you say: we have added a discussion to the text accordingly.

c) It may be convenient to define E = \rho \le T, as then it works when we take T = 0, which may be desirable.

Good point, done.

3.

a) In eq. 5, why omit r = 1? Why can't a solution be a single region?

There is not even such a limitation in the code! Again this is a slight hangover from an earlier implementation: we have removed the equation as it added little.

b) Above eq. 7, 'We can now expand on the previous definition of K'. The symbol K was used previously, but hardly defined. As a result, I found this discussion hard to follow.

The first K has been removed, and the text in-between clarified.

c) The 'expected Poisson likelihood-ratios between the signal-model under test and the background expectation' is used a proxy for power. As already mentioned, the paper would benefit from a clearer definition of what the authors mean by power.

i) The 'expected Poisson ...' under what model? Background only or new physics + background? How was this expectation computed numerically?

Given the negative results so far from the analyses being combined, our chosen definition of meta-analysis optimality is based on maximising model exclusions. Hence we compute the likelihood ratio of the SM Asimov dataset under the s+b and b models, controlling the Type 2 error rate. Maximising expected discovery sensitivity with signal Asimov data would be equally valid, corresponding to a controlled Type 1 rate.

ii) The connection between this proxy and the power should be commented on. Whilst it is clear that log likelihood between disjoint searches is additive, it is not so clear how power would 'add' or what the authors have in mind when searches are combined.

The composite LLR for each point is the thing we optimise; as mentioned above, this should be monotonic to the significance of exclusion, and hence exclusion-test statistical power, given some fairly reasonable approximations.

4.

a) What is the grey dashed line in Fig. 5?

It is the boundary of the efficiency-map tabulations. An explanation has been added to the caption.

b) I found the grey shading in Fig. 8 made the figure hard to read.

The distribution histograms have been redrawn with thicker lines, so they are more clearly visible now above the grey shading.

c) It is probably worth saying the distributions in Fig. 8 depend on the way in which the model parameters were sampled and specifying how it was done

The sampling and preselection of parameters for these studies is entirely determined by the (not very sophisticated) approach in the ATLAS Run 1 pMSSM-19 paper from which the parameter points were taken. We think it best left to the cited paper to explain the details.

We have added some text to specify how the ~20,000 points we analysed for each scenario were chosen from the larger ~100k ATLAS samples.

d) 'The expected transitions into the 95%-excluded p-value bin are less uniform than in the bino-LSP case, with only 12% of the least-excluded single-SR points expected to transition into the combined-SR exclusion bin' - I couldn't see how Fig. 10 showed this; should Fig. 11 be mentioned already here?

Yes, correct: these paragraphs were out of order. They have been rewritten to be more coherent.

e) I wasn't sure what 'least-excluded single-SR points' meant. Does it mean points in the p-value bin p > 0.95 with single SR?

Yes. Now made explicit.

f) 'This highlights how analysis combination makes limit-setting more robust against fluctuations in single analyses.' I didn't understand this conclusion. What is meant by robust here? The limits are robust against fluctuations in the sense that the type-1 error rate is controlled.

Apologies, this was not clear. The intention was to highlight that use of more signal regions means that not all eggs are placed in the expected-best basket, and observed upward fluctuations in expected-subleading SRs are automatically taken advantage of if able to include a larger subset.

g) 'The asymmetry of the distribution ...' I didn't fully grasp this: asymmetry about the mode? Why would we expect symmetry about the mode in the absence of the hereditary condition? I.e., why is it the hereditary condition that causes this asymmetry?

More than the asymmetry per se, which is influenced both by the typically non-zero information contributed by each SR and by the detail of the hereditary cutoff (giving a exponential suppression for large SR numbers), we simply wanted to highlight the rapid cutoff. The text has been rephrased.

h) The results in Fig. 13 are impressive! The final paragraph of this subsection that describes them beginning 'In a second step ...' seems quite short and unbalanced compared to the detail given in the other results sections.

Thank you. This section has now been substantially extended to discuss the contributions of the different analyses, and the impact of the SR-combination on final results.

Andrew Fowlie  on 2022-11-28  [id 3079]

(in reply to Humberto Reyes-González on 2022-11-23 [id 3062])

I have spent about 2.5 hours examining the changes and thinking about them. My first report made a number of minor suggestions. I would like to thank the authors for their patience working through my long report. For the most part, I was satisfied by the changes. I didn't see my comment 2.1 (a) addressed though.

My main concern was the lack of clarity over the notion of power used in the paper. This isn't particular to this paper: in my personal opinion, there is a lot of confusion and contradiction over the way 'power' is used in the collider pheno literature, compared to how it would be used (i.e., quite precisely) in a statistical context. The authors address this by adding a comment to the introduction and in the numbered points in sec. 3.2. Unfortunately, I don't think the issue here has been fully resolved. In particular,

i) In the collider literature and in the Asimov approach, we are often making the approximation that Median = expected, and using 'median' and 'expected' interchangeably. In this setting, I agree that the maximum expected significance may be found by maximzing the expected LLR.

ii) But this says nothing about power; it only says something about expected significance. I don't understand the 'power corresponds to ... ' part of the author's response or the 'This [power] is equivalent to maximising ...' part of point 3 in sec. 3.2. Why is maximzing expected significance equivalent to maximizing power?

As I commented in my first report, as far as I can tell, in this setting and under the Wald approximation, the power depends on the expectation of the LLR under the null *and* the alternative model (as well as a choice of fixed T1 error rate). Under each model, the test-statistic q = - 2 LLR follows a normal distribution with a mean and variance related by variance = 4. * |mean|. To find the power, you need to know the distribution of q under each model. Use the distribution under H0 to fix the T1 error. Use the distribution under H1 to compute the power at that fixed T1 error.

Perhaps there are some extra assumptions that the authors are making? The response hints at this (e.g., 'given some fairly reasonable approximations'; what approximations?). On the other hand, since it is common in the literature and since I expect power and expected significance under H1 to be quite closely related, it is reasonable to use it as a proxy, which is particularly useful in this setting since LLR is additive and so we have a sense in which power is additive which is required for the algorithm developed here. I think that would be perfectly reasonable so long as the logic is clarified.

Finally, there are typos in the changes in the text below eq. 3 (',,' and 'through by').

---

## Round 2 · Referee Report · Andrew Fowlie (Referee 1) · 2022-11-30

Strengths

See first report.

Weaknesses

See first report.

Report

I have spent about 2.5 hours examining the changes and thinking about them. My first report made a number of minor suggestions. I would like to thank the authors for their patience working through my long report. For the most part, I was satisfied by the changes. I didn't see my comment 2.1 (a) addressed though.

My main concern was the lack of clarity over the notion of power used in the paper. This isn't particular to this paper: in my personal opinion, there is a lot of confusion and contradiction over the way 'power' is used in the collider pheno literature, compared to how it would be used (i.e., quite precisely) in a statistical context. The authors address this by adding a comment to the introduction and in the numbered points in sec. 3.2. Unfortunately, I don't think the issue here has been fully resolved. In particular,

i) In the collider literature and in the Asimov approach, we are often making the approximation that Median = expected, and using 'median' and 'expected' interchangeably. In this setting, I agree that the maximum expected significance may be found by maximzing the expected LLR.

ii) But this says nothing about power; it only says something about expected significance. I don't understand the 'power corresponds to ... ' part of the author's response or the 'This [power] is equivalent to maximising ...' part of point 3 in sec. 3.2. Why is maximzing expected significance equivalent to maximizing power?

As I commented in my first report, as far as I can tell, in this setting and under the Wald approximation, the power depends on the expectation of the LLR under the null *and* the alternative model (as well as a choice of fixed T1 error rate). Under each model, the test-statistic q = - 2 LLR follows a normal distribution with a mean and variance related by variance = 4. * |mean|. To find the power, you need to know the distribution of q under each model. Use the distribution under H0 to fix the T1 error. Use the distribution under H1 to compute the power at that fixed T1 error.

Perhaps there are some extra assumptions that the authors are making? The response hints at this (e.g., 'given some fairly reasonable approximations'; what approximations?). On the other hand, since it is common in the literature and since I expect power and expected significance under H1 to be quite closely related, it is reasonable to use it as a proxy, which is particularly useful in this setting since LLR is additive and so we have a sense in which power is additive which is required for the algorithm developed here. I think that would be perfectly reasonable so long as the logic is clarified.

Finally, there are typos in the changes in the text below eq. 3 (',,' and 'through by').

Requested changes

1. Address outsanding item from first report (2.1 (a))
2. Clarify relation between power and expected LLR
3. Fix typos

  • validity: good
  • significance: good
  • originality: good
  • clarity: good
  • formatting: excellent
  • grammar: excellent

Author:  Humberto Reyes-González  on 2022-12-23  [id 3187]

(in reply to Report 1 by Andrew Fowlie on 2022-11-30)

Apologies for the missed point, and what we suspect is a wrong inference on our part about the connection between power and expected significance. We have addressed these points in the latest version, and respond to the review text below.

I didn't see my comment 2.1 (a) addressed though.

The corresponding paragraph has been rephrased.

We used a simplified model approach with the intention of obtaining robust conclusions about potential signal overlaps for arbitrary scenarios. In certain cases, such as simplified model base reinterpretation, the assessment of overlap between two analyses must be generalised for any scenario (within the convex hulls), since such approach doesn’t involve MC event generation and thus overlaps can’t be determined on the fly. Nonetheless, in the case of so-called full (MC-event based) recasting, we can certainly determine the orthogonality between LHC analyses for each specific scenario under consideration. In fact, we did this for Example 3 in the paper.

My main concern was the lack of clarity over the notion of power used in the paper. This isn't particular to this paper: in my personal opinion, there is a lot of confusion and contradiction over the way 'power' is used in the collider pheno literature, compared to how it would be used (i.e., quite precisely) in a statistical context. The authors address this by adding a comment to the introduction and in the numbered points in sec. 3.2. Unfortunately, I don't think the issue here has been fully resolved. In particular,

...

ii) But this says nothing about power; it only says something about expected significance. I don't understand the 'power corresponds to ... ' part of the author's response or the 'This [power] is equivalent to maximising ...' part of point 3 in sec. 3.2. Why is maximizing expected significance equivalent to maximizing power?

As I commented in my first report, as far as I can tell, in this setting and under the Wald approximation, the power depends on the expectation of the LLR under the null and the alternative model (as well as a choice of fixed T1 error rate). Under each model, the test-statistic q = - 2 LLR follows a normal distribution with a mean and variance related by variance = 4. * |mean|. To find the power, you need to know the distribution of q under each model. Use the distribution under H0 to fix the T1 error. Use the distribution under H1 to compute the power at that fixed T1 error.

You are quite right! Our intention was always to maximise the significance, and we had been using the word "power" informally to represent that... and then erroneously concluded that they were interchangeable here (we are not sure why, so thank you for insisting) and we could continue with that nomenclature.

You have clarified that it is indeed inaccurate, and probably confusing for more statistically minded readers, so have replaced the word "power" with "significance" or equivalent through the paper.

Finally, there are typos in the changes in the text below eq. 3 (',,' and 'through by').

Fixed! Thanks again.

---

## Round 3 · Referee Report · Andrew Fowlie (Referee 1) · 2023-1-2

Report

I would like to thank the authors for considering my comments and addressing my concerns. I now strongly recommend publication.

---

## Editorial Decision

published